

# No tropospheric ozone impact on the carbon uptake by a Belgian pine forest

Lore Verryckt[1], Maarten Op de Beeck[1], Johan Neirynck[2], Bert Gielen[1], Marilyn Roland[1], and Ivan A. Janssens[1]

[1]Department of Biology, University of Antwerp, Wilrijk, 2610, Belgium

[2]Research Institute for Nature and Forest, Geraardsbergen, 9500, Belgium

*Correspondence to:* L. Verryckt (lore.verryckt@uantwerpen.be)

**Abstract** High stomatal ozone ($O_3$) uptake has been shown to negatively affect crop yields and the growth of tree seedlings. However, little is known about the effect of $O_3$ on the carbon uptake by mature forest trees. This study investigated the effect of high $O_3$ events on gross primary production (GPP) for a Scots pine stand near Antwerp, Belgium over the period 1998-2013. Stomatal $O_3$ fluxes were modelled using in situ $O_3$ concentration measurements and a multiplicative stomatal model, which was parameterised and validated for this Scots pine stand. Ozone-induced GPP reduction is most likely to occur during or shortly after days with high stomatal $O_3$ uptake. Therefore, a GPP model parameterised for days with low stomatal $O_3$ uptake rates was used to simulate GPP during periods of high stomatal $O_3$ uptake. Eventual negative effects of high stomatal $O_3$ uptake on GPP would then result in an overestimation of GPP by the model during or after high stomatal $O_3$ uptake events. The $O_3$ effects on GPP were linked to AOT40 and $POD_y$. Although the critical levels for both indices were exceeded in every single year, no significant negative effects were found of ozone on GPP and no correlations between GPP residuals and AOT40 and $POD_y$ were found. Overall, we conclude that no $O_3$ effects were detected on the carbon uptake by this Scots pine stand.

## 1 Introduction

Tropospheric ozone ($O_3$) is a secondary air pollutant that has the potential to negatively affect vegetation, leading to reduced growth and carbon sequestration potential (ICP Vegetation, 2012; Middleton, 1956). Background concentrations of tropospheric $O_3$ have increased with 36 % since pre-industrial times (IPCC, 2001) and are projected to further increase considerably until about 2050 (IPCC, 2007). Depending on the scenarios, background $O_3$ levels might either increase or decrease after 2050 (IPCC, 2007).

In recent years, many studies have been conducted to investigate the mechanisms underlying the $O_3$ impacts on vegetation. Ozone reduces plant growth by altering photosynthetic rates, carbohydrate production, carbon sequestration, carbon allocation, and carbon translocation (Reich and Amundson, 1985;Andersen et al., 1997;Beedlow et al., 2004). Once $O_3$ enters the leaves through the stomata, it can affect plant growth by direct cellular damage (Mauzerall and Wang, 2001), leading to visible leaf injury and reduced leaf longevity (Noble and Jensen, 1980). In response to $O_3$, respiratory processes increase, which will also effect the tree's carbon balance (Darall, 1989). Skärby et al. (1987) proved that dark respiration of Scots pine shoots increased after long-term exposure to a low level of $O_3$. Protective responses, such as compensation (e. g. repair of injured tissue), avoidance



(e. g. stomatal closure), and tolerance (e. g. alteration of metabolic pathways), all consume carbon and, hence, resistance to $O_3$ damage costs energy. The size of this cost affects the amount of carbon remaining to support growth (Skärby et al., 1998).

To assess the impact of $O_3$, several indices have been created, e. g. AOT40 (ppb h), the cumulated $O_3$ concentration in excess of a threshold of 40 ppb, and $POD_y$, the accumulated $O_3$ flux above a flux threshold y (nmol m$^{-2}$ s$^{-1}$). Critical levels, quantitative estimates of exposure to $O_3$ above which direct adverse effects may occur (CLRTAP, 2015), have been determined for these indices based on $O_3$ dose-response relationships from fumigation experiments with enhanced $O_3$ concentrations (Karlsson et al., 2004). The magnitude of the $O_3$ impact on plants depends on the intensity of $O_3$ exposure, environmental factors influencing both plant photosynthesis and the $O_3$ flux to plant surfaces, and plant species-specific defensive mechanisms (Musselman and Massman, 1999). Because of the variable plant responses to similar $O_3$ concentrations, the question arises whether widely applicable tolerable limits of $O_3$ concentration exist (Skärby et al., 1998).

While high stomatal $O_3$ fluxes have been shown to affect crop yields and tree seedlings, it is not sure whether $O_3$ uptake or $O_3$ flux also negatively affects the carbon uptake by mature forest trees. Many studies determined the effect of $O_3$ on seedlings and young trees (Buker et al., 2015), but little is known about the effect on mature trees. When scaling up the results from seedlings to mature trees the resulting data should be viewed with caution, due to differences in energy budgets, canopy:root balances and architecture and carbon allocation patterns (McLaughlin et al., 2007;Chappelka and Samuelson, 1998). In addition to the uncertainties related with the up-scaling from seedlings to mature trees, data from controlled experiments should also be used with caution, because trees can react differently in field conditions (Skärby et al., 1998). The effect of $O_3$ uptake on carbon uptake under ambient $O_3$ concentrations by trees has hardly been studied in situ. Some studies showed reductions in plant growth due to stomatal $O_3$ uptake (Zapletal et al., 2011;Fares et al., 2013;Yue and Unger, 2013), while other studies did not show any effect (Zona et al., 2014;Samuelson, 1994). Whether or not an effect of stomatal $O_3$ uptake was found was species- and site- specific, and there is a clear need for more studies investigating the effect of $O_3$ on carbon uptake by mature trees in the field (Chappelka and Samuelson, 1998).

Here we investigate the effect of high $O_3$ events on gross primary production (GPP) for a Scots pine stand in Flanders, Belgium. At current ambient $O_3$ levels, critical levels for both AOT40 and $POD_1$ are already being exceeded in this Scots pine stand (Neirynck et al., 2012). This indicates that even at current ambient $O_3$ levels tree productivity might be affected. Ozone-induced GPP reduction is most likely to occur during or shortly after days with high stomatal $O_3$ uptake. An effect of stomatal $O_3$ uptake on GPP can be detected when a GPP model parameterised for days with low stomatal $O_3$ uptake rates is extrapolated to high stomatal $O_3$ uptake events – i. e., days where an effect on GPP is assumed - and the model overestimates GPP during these events. This study therefore tests the hypothesis that GPP of the studied pine forest is reduced during or shortly after high stomatal $O_3$ uptake events.

## 2 Materials and methods



### 2.1 Study area

The study area consisted of a 2-ha Scots pine stand in a 150-ha coniferous/deciduous forest named 'De Inslag', situated in Brasschaat (+51° 18' 33'' N, +04° 31' 14'' E), northeast of the Antwerp agglomeration and east-northeast of the Antwerp harbour (Neirynck et al., 2008). The site has a temperate maritime climate with a mean
annual temperature of 11 °C and a mean annual precipitation of 830 mm (Neirynck et al., 2008).

The soil has been classified as Albic Hypoluvic Arenosol (Gielen et al., 2011), a moderately wet sandy soil with a distinct humus and/or iron B-horizon (Janssens et al., 1999). The sandy layer overlays a clay layer which is situated at a depth of 0.7 - 2 m. As a result of the poor drainage groundwater depth is typically high, fluctuating between 0.5 and 2 m (Carrara et al., 2003).

The trees were planted in 1929 (Neirynck et al., 2008). In 1995, tree density amounted to 542 trees ha$^{-1}$. In the autumn of 1999, the forest was thinned, which resulted in 376 trees ha$^{-1}$ in 2001. With a peak in leaf area index (LAI) of $1.3 \pm 0.5$ m$^2$ m$^{-2}$ in 2007 (Op de Beeck et al., 2010) and an average LAI of $1.2 \pm 0.5$ m$^2$ m$^{-2}$ in the period 1998-2007, the stand canopy is very sparse. Only two needle-age classes are present: current-year needles and one-year-old needles (Op de Beeck et al., 2010).

### 2.2 Measurements: meteorology, $O_3$, GPP, and LAI

In this study, continuous measurements over the period 1998 - 2013 were used, excluding 1999 and 2003 due to poor data quality or coverage. Different meteorological variables were measured on a tower with a height of 41 m, set up in 1996 (Gielen et al., 2013). A sonic anemometer (Model Solent 1012R2, Gill Instruments, Lymington, UK) measures the wind velocity (WV; m s$^{-1}$). Meteorological data include vertical profiles of air temperature ($T_{air}$;
°C) and humidity (RH; (%) (HMP 230 dew point transmitter and PT100, Vaisala, Finland) in aspirated radiation shields at 2, 24 and 40 m height. Wind speed measurements (LISA cup anemometer, Siggelkow GMBH, Germany) are conducted at 24, 32 and 40 m height. At the top of the tower, ingoing and outgoing short-wave and long-wave radiation are measured by a CNR1-radiometer (pyranometer/pyrgeometer, Kipp and Zonen, the Netherlands) and a CMP6-pyranometer (Kipp and Zonen, the Netherlands). A wind vane (potentiometer W200P, Campbell, UK) is
mounted on a tower rail. Rainfall is registered by a tipping bucket rain gauge (NINA precipitation pulse transmitter, Siggelkow GMBH, Germany). Both $T_{air}$ and RH are used to calculate vapour pressure deficit (VPD; kPa). Soil temperature ($T_{soil}$; °C) is measured at 9 cm below the soil surface with temperature probes (Didcot DPS-404, UK). Soil water content (SWC; m$^3$ m$^{-3}$) was measured at 25 cm below the soil surface using a TDR (Time Domain Reflectometer) sensor at three days to biweekly resolution and subsequently interpolated to obtain daily estimates,
taking into account water inputs via precipitation (Gielen et al., 2010). Soil water potential (SWP; MPa) was derived from SWC measurements (m$^3$ m$^{-3}$) with the model of van Genuchten (van Genuchten, 1980). All climatic variables were measured every 10 seconds and half hourly means were stored on a data logger (Campbell CR1000, UK) in an air-conditioned cabinet adjacent to the tower. Data gaps were filled with data from nearby weather stations.

Vertical profiles of $O_3$ concentrations are being measured at two inlets above the canopy (at 24 and 40 m) using an UV Photometric Analyzer (model TEI 49I, Thermo Environmental Instruments). $CO_2$ concentrations (ppm) are



measured using an infrared gas analyser (IRGA) (Model LI-6262, LI-COR Inc., Lincoln, NE, USA). The vertical $CO_2$ flux between the forest and the atmosphere, net ecosystem exchange (NEE), was measured with the eddy covariance technique following standard data quality procedures (Carrara et al., 2003;Gielen et al., 2013;Carrara

et al., 2004). Gross primary production ($\mu$mol C m$^{-2}$ s$^{-1}$) is derived from NEE data, by subtracting the modelled total (autotrophic and heterotrophic) ecosystem respiration from the measured NEE. The ecosystem respiration or total carbon loss is modelled with standardised algorithms as presented in Falge et al. (2001).

The LAI time series for the Brasschaat forest was reconstructed based on the historical data. The general approach was to use the fragmentary LAImax data that were measured by Gond et al (1999) in 1997, by Konôpka et al.

(2005) in 2003 and by Op de Beeck et al. (2010) in 2007. The latter was done with hemispherical pictures while the first in 1997 and 2003 were done with the LAI-2050 instrument (LI-COR, Lincoln, Nebraska, USA). To assure consistency across the time series, measurements were corrected for clumping by using the factor 0.83 (Jonckheere et al., 2005). All LAImax measurements were interpolated lineairly to derive LAImax values for the missing years. The thinning event in 1999 was accounted for by subtracting the removed leaf biomass by using the allometric

relations from Yuste et al. (2005) and specific leaf are measurements from Op de Beeck et al. (2010). The seasonal pattern of the Op de Beeck et al. (2010) measurements was used and kept constant over the entire time series.

### 2.3 Stomatal conductance measurements

During the summers of 2007 and 2013, stomatal conductance to $H_2O$ ($g_{st, H2O}$) of Scots pine needles was measured at the site. These $g_{st,H2O}$ measurements were based on gas exchange measurements (photosynthesis and

transpiration), which were carried out with a leaf cuvette connected to an IRGA (LI-6400, LI-COR, Lincoln, Nebraska, USA).

Stomatal measurements were carried out in highly different environmental conditions during 2007 (cold and wet summer) and 2013 (warm and dry summer). Diurnal stomatal measurements and stomatal responses to PAR, $T_{air}$, and VPD, both on one-year-old needles and on current-year needles, were carried out during the summers of 2007

(Op de Beeck et al., 2010) and 2013. All measurements in 2007 were carried out on the needles of the two trees closest to the tower, both in the lower and the upper canopy. In 2013 only the tree closest to the tower was accessible for measurements. The total number of sets of needles on which measurements were carried out, amounts to 10 in 2007 and 16 in 2013. The LI-6400 instrument calculated the $g_{st,H2O}$ assuming the whole area of the cuvette (2x3 cm) was covered with the needles. To obtain the $g_{st}$ to $O_3$ (from here on referred to as '$g_{st}$') of the

six or eight needles in the cuvette, two corrections needed to be made: one for the needle area, because needles did not completely cover the area of the cuvette, and a second one for converting the data from $g_{st, H2O}$ to $g_{st}$.

The width of the needles was measured at 40 x magnification under a binocular microscope (M3 Wild, Wild Heerbrugg, Gais, Switzerland) in combination with an ocular equipped with a reticule (Leitz, Wetzlar, Germany, periplan, GW 10xm). The width per needle was measured at three places: at the top, in the middle, and at the base.

An average of those three measurements was multiplied by the length of the needle inside the cuvette, 3 cm. After measuring the needle area with a microscope, the $g_{st,H2O}$ data were corrected for the lower needle area. These data



were multiplied by 0.61, which is the ratio of the molecular diffusivities of water vapour and $O_3$ in the air, to convert from $g_{st,H2O}$ to $g_{st}$.

**2.4 Multiplicative stomatal model: description**

Stomatal conductance was modelled using the multiplicative $g_{st}$ model, first described by Jarvis (1976). The model has been developed to calculate species-specific $g_{st}$ according to phenology and environmental conditions (Emberson et al., 2000) and is described in detail in Appendix A.

We modified the model to make it more applicable for Scots pine. In this modified model (Eq. 1) PAR, $T_{air}$, VPD, and SWP influence the range between $g_{max}$ and $g_{min}$ instead of $g_{max}$ and zero. This modification was needed, because

in the Brasschaat pine forest stomata never completely close, hence $g_{st}$ is never zero (Op de Beeck et al., 2010).

$$g_{st} = g_{max} * f_{phen} * (f_{min} + (1 - f_{min}) * (f_{PAR} * f_T * f_{VPD} * f_{SWP}) \qquad (1)$$

Here $g_{st}$ is the stomatal conductance to $O_3$ and $g_{max}$ is the maximal stomatal conductance to $O_3$. The functions $f_{PHEN}$, $f_{PAR}$, $f_T$, $f_{VPD}$, and $f_{SWP}$ represent the modification of $g_{max}$ by, respectively, phenology, PAR, $T_{air}$, VPD, and SWP. The function $f_{min}$ is the ratio of $g_{min}$ and $g_{max}$ where $g_{min}$ is the minimal stomatal conductance to $O_3$ (see Appendix

A for more detailed information). Impaired stomatal aperture mechanisms (stomatal sluggishness) due to ozone exposure (Paoletti and Grulke, 2010) were not included in the model development.

**2.4.1 Multiplicative stomatal model: parameterisation and validation**

For the optimisation of the parameters of the different functions in the model, we assumed that the phenology function was 1. This was deemed a fair assumption, because $g_{st,H2O}$ was measured on mature needles in the summer

(July/August 2007 and 2013), in the middle of the growing season.

The data set, including measured $g_{st}$, PAR, $T_{air}$, VPD, and SWP, was split into two subsets by grouping odd and even rows for data sorted by PAR. One set was then used for parameterisation, the other for validation. The stomatal model was parameterised using the computer program Matlab (version 2013a). The optimisation of all parameters was done with the function 'lsqcurvefit' in Matlab. It finds the best parameter values, starting with an

initial value, to best fit the function of the stomatal model to measured $g_{st}$ and can thus be used to fit a nonlinear function with more than two independent variables. All parameters of $f_{PAR}$, $f_{Tair}$, $f_{VPD}$, and $f_{SWP}$ were optimised separately, with initial values that were estimated visually from plots of the functions to the dataset.

**2.4.2 Multiplicative stomatal model: model evaluation**

The parameterised multiplicative stomatal model was then tested against the validation dataset. Measured $g_{st}$ values

were plotted against the modelled $g_{st}$ values. A linear function $y = ax + b$ was fitted, where 'a' should be not significantly different from one ($p > 0.05$) and 'b' should be not significantly different from zero ($p > 0.05$) for both parameterisation and validation dataset. We evaluated the model performance with the following statistics: the coefficient of determination or R squared ($R^2$) as a goodness-of-fit measure and error-based measures including





mean bias (MB), relative mean error (RME), Willmott's index of agreement (d), model efficiency (ME), root mean squared error (RMSE), and its systematic (RMSE$_s$) and unsystematic component (RMSE$_u$). In Appendix B these error-based statistics are explained.

The measured g$_{st}$ was plotted in function of the different input variables (PAR, T$_{air}$, VPD, and SWP) and the boundary function of each plot was fitted. This was done in order to test how well the obtained parameter values were estimated in function of the measured g$_{st}$.

**2.5 Canopy model**

We applied a canopy model to scale up g$_{st}$, measured at leaf level, to the canopy level. The canopy model consists of different horizontal leaf layers and includes a radiation transfer model (Goudriaan, 1977), a solar elevation model (Campbell and Norman, 1998) and the modified multiplicative stomatal model (Emberson et al., 2000). The model is described in detail in Appendix C.

The model calculates half-hourly totals of the total, stomatal, and non-stomatal O$_3$ fluxes based on the following input variables: day of year, hour, Rg, T$_{air}$, VPD, SWP, O$_3$ concentration above the canopy (24m), LAI, and friction velocity u*. The total O$_3$ flux (nmol m$^{-2}$ s$^{-1}$) for the whole canopy was the product of O$_3$ concentrations (ppb) and g$_{tot}$ (mol (m² ground area$^{-1}$) s$^{-1}$) (Musselman and Massman, 1999). This last parameter was calculated with an electrical model (Eq. 2).

$$g_{tot} = (\frac{1}{g_{aero}} + \frac{1}{g_{bl}} + \frac{1}{g_{can}})^{-1} \qquad (2)$$

where g$_{tot}$ is the total conductance to O$_3$ (mol (m² ground area$^{-1}$) s$^{-1}$); g$_{aero}$ is the aerodynamic conductance and is set to 1; g$_{bl}$ is the boundary layer conductance to O$_3$; g$_{can}$ is the canopy conductance.

The boundary layer conductance to O$_3$ was calculated with the following formula (Baldocchi et al., 1987):

$$g_{bl} = \frac{K*u^*}{2*\frac{Sc^{\frac{2}{3}}}{Pr}} * 44.64 \qquad (3)$$

where K is the von Karman constant (0.43); u* (m s$^{-1}$) is the friction velocity, which is derived from the measured momentum fluxes; Sc is the Schmidt number (1.07 for O$_3$); Pr is the Prandtl number (0.72 for O$_3$); 44.64 mol m$^{-3}$ is the molar density of air, and is applied for converting the unit of g$_{bl}$ from m s$^{-1}$ to mol m$^{-2}$ s$^{-1}$.

The canopy conductance consisted of a stomatal and a non-stomatal component. Since the stomatal component varies throughout the canopy, the canopy was divided into eight sublayers so that the leaves were evenly distributed 200 between the horizontal layers. Dividing the canopy into sufficient sublayers was necessary in order to model fluxes well. Eight sublayers were considered to be sufficient, as indicated in a sensitivity test with more and less sublayers. For each leaf layer, the model calculates g$_{st}$ for sunlit and shaded needles, taking the solar elevation angle into account. Non-stomatal conductance was assumed to be constant over the canopy and was set at 0.16. This value was derived from long-term O$_3$ flux measurements in Brasschaat (Neirynck et al., 2012).





The stomatal and non-stomatal $O_3$ fluxes (nmol m$^{-2}$ s$^{-1}$) were calculated by multiplying the proportion of $g_{st}$ and $g_{ns}$ of the canopy per ground area with the $O_3$ concentration.

These obtained half-hourly fluxes were aggregated to daily fluxes. These daily fluxes were averaged in order to know the average daily $O_3$ uptake by the canopy for the different years. The ratio $F_{st}/F_{tot}$ was calculated and this gives an indication of the contribution of the stomatal $O_3$ flux to the total $O_3$ flux.

**2.6 Ozone effects**

A feed-forward back propagation Artificial Neural Network (ANN) in Matlab (Matlab Matworks R2013a, The MathWorks Inc., Natick, Massachusetts, USA) was used to simulate GPP of the Scots pine forest. The ANN, which contained 10 nodes organised in 1 layer, was trained with 70% data random selected data measured data and validated based on the remaining 30% of data set (R²=0.72). The daily GPP data of the growing seasons

between 1998-2013, except 1999 and 2003, were used as dependent target variable, whereas year, day of year, $T_{min}$, $T_{max}$, $T_{mean}$, average VPD, SWC, Rg, average $T_{soil}$, and average WV of these periods were used as independent input variables. Daily totals of the variables were used, with the exception of VPD, $T_{soil}$, and WV for which daily averaged values were used.

To obtain an $O_3$-damage free GPP model, data from days for which an $O_3$ effect was expected were removed from

the dataset. These were the days with stomatal $O_3$ uptake values in the upper 2%, 5 %, and 10% of stomatal $O_3$ flux. As the results of a 2% and 10% cut-off were equal to a 5% cut-off, we report only results of a 5% cut-off. With 2/3 of the data from days with low stomatal $O_3$ uptake, the artificial neural network was trained. The other 1/3 was used to test the model. This $O_3$-damage free GPP model was then run with all data. The absolute and relative differences in GPP accumulated over the growing season between EC-derived and modelled values were

calculated, to investigate whether or not there was a reduction of GPP.

The relation between the residuals of total GPP and both AOT40, POD$_1$ and POD$_2$ was examined. Therefore, a linear fit between the residuals and the indices was made. A significant negative correlation would exist if the slope is significant different from 0 ($p < 0.05$) and intercept is not significant different from 0 ($p > 0.05$). These yearly GPP residuals were also plotted to the stomatal $O_3$ flux to investigate their relation and a linear fit was made

of which the significance was tested. If GPP was increasingly overestimated in the presence of higher stomatal $O_3$ fluxes, this would indicate a deleterious $O_3$ effect.

Ozone effects possibly appear and last during a period of several days after the $O_3$ peaks, and as a result they will not be detected in the above analyses. Due to these possibly lag effects of $O_3$, the above analyses were repeated, but now excluding the days with high stomatal $O_3$ uptake along with the two subsequent days removed from the

training and testing datasets.





## 3 Results

### 3.1 Measurements: meteorology, GPP, and LAI

A fingerprint of the multi-annual average diel and seasonal pattern in the measured data is shown in Fig. 1. This figure gives a good overview of how meteorology and GPP typically changed over time in this forest; interannual
anomalies from the average patterns can be found in Fig. S1. Distinct daily and seasonal patterns can be observed for $T_{air}$, $R_g$, and VPD, reaching highest values in summer, in the afternoon. Similar patterns can also be observed in GPP, which basically follows the pattern of Rg. As seen on Fig. 1, the photosynthetic period extends, on average, from day of year 115 (end of April) till day of year 300 (end of October). The precipitation and SWP time series are provided in Fig. 2, while changes in LAI over time are shown in Fig. 3The yearly maximum LAI ranged from
1.4 to 1.9 $m^2$ $m^{-2}$. The thinning of the forest in 1999 can clearly be observed in the LAI pattern. After the thinning, the canopy never fully closed.

### 3.2 Multiplicative stomatal model and simulated O₃ fluxes

The best fitting parameter values for the multiplicative stomatal model are presented in Table 1 and different statistics to evaluate the model performance are presented in Table 2. For the parameterisation dataset, the
measured data were fitted against modelled $g_{st}$ and plotted in Fig. 4. The slope of the linear fit was not significantly different from 1 ($p > 0.05$) and the intercept was not significantly different from 0 ($p > 0.05$). Model evaluation for the validation dataset was equally good as for the parameterisation dataset (Table 2). Also in the linear fit for the validation set (Fig. 4, B), the slope was not significantly different from 1 ($p > 0.05$) and the intercept was not significantly different from 0 ($p > 0.05$).

In Fig. 5 the measured $g_{st}$ was plotted against the different model input variables: PAR, $T_{air}$, VPD, and SWP, and for each plot the boundary function was fitted.

The average daily O₃ fluxes for the different years are presented in Fig. S2. The daily total $F_{st}$ ranges from 1.42 to 2.00 nmol O₃ $m^{-2}$ $day^{-1}$. In 2011 the daily total $F_{st}$ was the lowest, while in 2002 the highest stomatal flux was reached. The annual average ratio $F_{st}/F_{tot}$ varied between 24-28 % (Fig. S2). We observed the lowest ratios in the
beginning and at the end of the growing season. Above-average ratios were observed at the peak of the growing season.

### 3.3 Ozone effects on GPP

Total GPP (mol C $m^{-2}$ $day^{-1}$) was calculated for days with low stomatal O₃ uptake, high stomatal O₃ uptake and for the entire growing season, using both the EC-derived GPP data and the modelled GPP data (Table 3). For days with
low stomatal O₃ uptake, the average daily total GPP was 0.48 mol C $m^{-2}$ $day^{-1}$, and the models reproduced GPP very well (Table 3). When we calculated total GPP for days with high stomatal O₃ uptake, the EC-derived fluxes were much higher than for the days with low stomatal O₃ uptake. This was probably due to the higher irradiation that typically occurs during high O₃ events and stimulates GPP. The higher GPP, however, also suggests that negative O₃ effects on GPP were highly unlikely. This is exacerbated by the fact that our models almost





consistently underestimate GPP during high $O_3$ events (Table 3), whereas we hypothesised the exact opposite, namely that the models would overestimate GPP during these events because they were parameterised for low $O_3$ days. We also observed no differences between both models, suggesting no lagged $O_3$ effects on GPP (Table 3).

A weak, negative correlation between total GPP residuals and $F_{st}$ exists for the GPP model trained without days with high stomatal $O_3$ uptake (Fig. 7, A), while a small positive correlation is shown for the GPP model which

tested for lag effects of $O_3$ (Fig. 7, B). However, these differences were not statistically significant at $p<0.05$. For both models, correlations between total GPP residuals and AOT40, and between total GPP residuals and both $POD_1$ and $POD_2$ existed. These correlations were also not statistically significant at $p<0.05$ (Fig. 7, C, D, E, F, G, and H).

## 4 Discussion

### 280 4.1 Multiplicative stomatal model

All statistics shown in Table 2 clearly indicated that the fitted multiplicative stomatal model performed well. For both parameterisation and validation datasets, the model explained 72 % of the variance in $g_{st}$. For both datasets, slope and intercept of the linear regression lines of measured versus modelled $g_{st}$ were not significantly different from 1 and 0, respectively (Fig. 4). Moreover, the model efficiency (ME in Table 2) of 0.72 and the Wilmott's

index (d) close to 1 both indicate that the modelled values matched the measured values well. A good model provides low root-mean-square error (RMSE), while the systematic component ($RMSE_s$) should approach zero and the unsystematic component ($RMSE_u$) should approach RMSE (Willmott et al., 1985), which was the case for this model. Low mean bias (MB) and low mean relative error (MRE) further indicated very good performance. The good performance of the model can also be observed in Fig. 5, in which the boundary lines represented the

response of $g_{st}$ to the independent variables when other variables were not limiting (Martin et al., 1997). The boundary lines fitted close to the data points, which is an indication of a good model, because the multiplicative stomatal model is based on the assumption that the variables act more or less multiplicatively and independently from each other (Grüters et al., 1995).

Multiplicative stomatal models based on Jarvis (1976) have been parameterised earlier for generic Scots pine

forests in Europe (Mills et al., 2011;Buker et al., 2015) and used to estimate critical levels for this species. However, the empirical the dose-response relationship for Scots pine is based on only one two-year fumigation study on small seedlings and, therefore, high uncertainty exists in the modelled $O_3$ impact on Scots pine growth.

The parameterisation of Mills et al. (2011) and Büker et al. (2015) differ from that of this study in a number of parameters. First, the needles of the Scots pine stand in Brasschaat had a higher night-time $g_{st}$ ($g_{min}$) and will

therefore take up more $O_3$ at night. Maximal $g_{st}$, in contrast, is lower in Brasschaat than estimated for other Scots pine forests, implying that during episodes of high $O_3$ concentrations, the Brasschaat site is unlikely to take up very high amounts of $O_3$. This may have contributed to the absence of a clear $O_3$ response at our study site. Also the Scots pine stand in Brasschaat is less sensitive to drought stress than the generic model, due to a higher $VPD_{max}$ and a wider SWP range. The wider SWP range is mainly due to a clearly lower $SWP_{max}$. These differences between

the parameter values and, hence, in $g_{st}$ for generic Scots pine forests and for the Scots pine stand in Brasschaat will





lead to different critical levels and under- or overestimation of possible $O_3$ damage. Species-specific parameterisation is important, but site-specific parameterisation is clearly important as well.

### 4.2 Stomatal $O_3$ fluxes at canopy scale

The stomatal $O_3$ flux contributed on average for 26 % to the total $O_3$ flux over the study period (Fig. S2). This

fraction is similar to the 21 % stomatal $O_3$ flux in a Danish Norway spruce stand (Mikkelsen et al., 2004) and the 30 % stomatal $O_3$ flux in *Quercus ilex* in Italy (Vitale et al., 2005;Gerosa et al., 2005). Cieslik (2004) showed that in Southern Europe stomatal $O_3$ flux of different vegetation types, such as pine forest and Mediterranean shrubs, is typically less than 50 % of the total $O_3$ flux. A five-year study on a Mediterranean *Pinus ponderosa* stand showed a stomatal $O_3$ flux contribution of 57 % (Fares et al., 2010). Clearly species- and site-specific differences such as

tree age or micro-climate are introducing large variability in stomatal $O_3$ uptake (Neirynck et al., 2012).

The low relative stomatal $O_3$ flux in the Scots pine stand in Brasschaat could be the result of the sparse canopy with low LAI. Although no relation between stomatal $O_3$ flux and LAI was found in a previous site study on this site (Neirynck et al., 2012), interannual and seasonal variation in LAI is very small, rendering such a correlation analysis very difficult.

### 4.3 Ozone effects on GPP

Although our models reproduced GPP very well, we did not observe immediate or lagged effects of high stomatal $O_3$ uptake on GPP (Table 3; Fig. 7, A, B). Some earlier studies have investigated the effect of $O_3$ on forest carbon uptake. Cumulative stomatal uptake of 27 mmol $m^{-2}$ over the growing season did not result in any visible damage or a reduction in NEE of a poplar plantation in Belgium (Zona et al., 2014). Zapletal (2011), on the other hand,

reported that $CO_2$ uptake of a Norway spruce forest in the Czech Republic increased with increasing stomatal $O_3$ flux, followed by a sudden decrease in $CO_2$ uptake, suggesting that an $O_3$ flux threshold exists. Fares (2013) showed a negative correlation between GPP and $O_3$ uptake at two Mediterranean ecosystems (a forest dominated by *Pinus ponderosa* in California, USA and an orchard site of *Citrus sinensis* cultivated in California, USA). A GPP reduction of 1-16% in response to $O_3$ uptake under ambient $O_3$ concentrations of 30-50 ppb was determined

across vegetation types and environmental conditions in the United States by Yue and Unger (2013). The magnitude of reduction depended on the sensitivity to $O_3$ of the species and on the biome types.

AOT40 is, at present, the European standard for forest protection (EEA, 2014), with a critical level of 5000 ppb h, equivalent to a growth reduction of 5 % (Mills et al., 2011). In this study on Scots pine in Brasschaat, this value was far exceeded in all years (Fig. 7), yet no negative effect on GPP was observed in years with higher AOT40

values. Particularly noteworthy is the extreme high AOT40-value of 2006, which was due to the high $O_3$ concentrations during that year, which, nevertheless, did not result in GPP reductions (Table 3).

$POD_y$ is considered a more appropriate index for potential $O_3$ damage because it considers $O_3$ flux. The critical level of $POD_1$ is species-specific; a critical level of 8 mmol $m^{-2}$ with 2 % growth reduction is used for Norway spruce and a critical level of 4 mmol $m^{-2}$ with 4 % growth reduction is used for birch and beech (Mills et al., 2011).

A critical level for Scots pine has not yet been determined and therefore the value of 8 mmol $m^{-2}$ for Norway





spruce is often adopted as critical level for Scots pine. During this study, this critical level was exceeded every single year, and again no negative correlation between total GPP residuals and $POD_1$ was observed. In comparison to the AOT40 level, 2006 was not the year with the highest $POD_1$. This difference between AOT40 and $POD_1$ during 2006 was due to stomatal closure; during high $O_3$ concentration events, $g_{st}$ was rather low (Fig. 6). $POD_1$

was highest in the year 2002, when $O_3$ concentrations were relatively low, but $g_{st}$ was high. The low $O_3$ concentrations explain the lower AOT40 for 2002.

Notwithstanding the absence of a statistically significant negative correlation between GPP residuals and both AOT40 and $POD_y$, critical levels for both AOT40 and $POD_y$ were exceeded every single year. AOT40 is based on $O_3$ concentration and these concentration-based indices have been shown to be weaker indicators for $O_3$ damage

than flux-based indices (Karlsson et al., 2007;Simpson et al., 2007). The critical level of $POD_y$ for Scots pine was adopted from the critical level for Norway spruce (Mills et al., 2011). Possibly this critical level is too low for Scots pine. As shown by Reich (1987), pines are less sensitive to $O_3$ compared to hardwoods and crops. This supports the idea of a too low critical level.

Overall, no significant $O_3$ effects on GPP accumulated over the growing season were found. Although no

significant $O_3$ effects on GPP were found in this study, it is still possible that $O_3$ negatively affected this Scots pine stand in Brasschaat. Stomatal $O_3$ uptake was linked to reductions in GPP only. As already stated in the introduction, protective responses such as compensation and enhanced tolerance occur in trees (Skärby et al., 1998). It is likely that trees at our study site were able to fully detoxify the incorporated $O_3$. As a result, no $O_3$ effects on carbon uptake were detectable. However, this protection may have come at a respiratory cost, which may have reduced

the NPP/GPP ratio of this forest. The NPP/GPP ratio of our study site was very low (Gielen et al., 2013). In addition to the poor nutrient status (limitation by P and Mg, extremely high N deposition; (Neirynck et al., 2008)), $O_3$ uptake may partly be responsible. This can, however, not be tested because pine forest NPP data were not available at annual timescale.

### 5 Summary

We parameterised a multiplicative stomatal model for a Scots pine stand in Brasschaat. This species- and site-specific parameterised model performed very well. With this model, stomatal $O_3$ fluxes were calculated and used to test for $O_3$ effects on GPP. Only very small reductions in growing season GPP were calculated. Although critical levels for AOT40 and $POD_y$ were exceeded in every single year, no significant negative correlations between total GPP residuals and stomatal $O_3$ flux, AOT40, and $POD_y$ were found. In general, we can thus conclude that no $O_3$

effects were detected on the carbon uptake by the Scots pine stand in Brasschaat.




**Appendix A The multiplicative stomatal model**

*In this work the multiplicative stomatal model described by Jarvis (1976) is modified specific for the Scots pine stand in Brasschaat. The basic model is explained below.*

The multiplicative stomatal model is described by Jarvis (1976) and modified by Emberson (2000);

$$g_{st} = g_{max} * f_{phen} * \max(f_{min}; f_{PAR} * f_T * f_{VPD} * f_{SWP}) \tag{A1}$$

where $g_{max}$ is the species-specific maximum stomatal conductance to $O_3$ (mmol m$^{-2}$ s$^{-1}$) expressed on a total leaf surface area basis. The other parameters are expressed in relative terms as a proportion of $g_{max}$;

$f_{min}$ is the ratio of $g_{min}$ to $g_{max}$; where $g_{min}$ is the minimal stomatal conductance that occurs during daylight

period;

$f_{phen}$ represents the modification of $g_{max}$ due to phenological changes;

$f_{PAR}$ represents the modification of $g_{max}$ by photosynthetically active radiation (PAR);

$f_T$ represents the modification of $g_{max}$ by air temperature ($T_{air}$);

$f_{VPD}$ represents the modification of $g_{max}$ by vapour pressure deficit (VPD);

$f_{SWP}$ represents the modification of $g_{max}$ by soil water potential (SWP).

Phenology modifies $g_{max}$ because of the variation in $g_{st}$ due to differences in needle age. The function can be modelled as follows:

$$\tag{A2}$$

if $SGS \leq doy \leq (SGS + c)$, then $f_{PHEN} = f_{min} + (1 - f_{min}) * (1 - b) * \left(\frac{doy - SGS}{c}\right) + b$

if $SGS + c \leq doy \leq EGS - d$, then $f_{PHEN} = f_{min} + (1 - f_{min}) * 1$

if $EGS - d \leq doy \leq EGS$, then $f_{PHEN} = f_{min} + (1 - f_{min}) * (1 - b) * \left(\frac{EGS - doy}{d}\right) + b$

where SGS is the start of the growing season;

EGS is the end of the growing season;





b, c, and d are species-specific parameters representing the minimum of $f_{PHEN}$, the number of days for

$f_{PHEN}$ to reach its maximum and the number of days during the decline of $f_{PHEN}$ for the minimum to reach

again, assuming linear increase and decrease at the start and end of the growing season.

The stomatal response PAR is described by a rectangular hyperbola, where $a_{PAR}$ is a species-specific

parameter determining the shape of the hyperbola (Emberson et al., 2000);

$$f_{PAR} = 1 - \exp(-a_{PAR} * PAR) \qquad (A3)$$

The stomatal response to $T_{air}$ is given by a parabolic function, where $T_{min}$ is the minimum temperature

at which stomatal opening occurs, and $T_{opt}$ is the optimum temperature of stomatal opening (Emberson

et al., 2000);

$$f_T = \max\left(0; 1 - \frac{(T - T_{opt})^2}{(T_{opt} - T_{min})^2}\right) \qquad (A4)$$

The stomatal response to VPD is described by the following relationship, where $VPD_{min}$ is a threshold

for minimal stomatal opening, and $VPD_{max}$ is a threshold for full stomatal opening (Emberson et al.,

2000);

$$f_{VPD} = \min\left(1; \max\left(0; \frac{VPD_{min} - VPD}{VPD_{min} - VPD_{max}}\right)\right) \qquad (A5)$$

The stomatal response to SWP is described by the following relationship, where $SWP_{min}$ is a threshold

for minimal stomatal opening, and $SWP_{max}$ is a threshold for full stomatal opening (Emberson et al.,

410     2000);

$$f_{SWP} = \min\left(1; \max\left(0; \frac{SWP_{min} - SWP}{SWP_{min} - SWP_{max}}\right)\right) \qquad (A6)$$



**Appendix B Statistics of model performance**

*In order to test how well the modified stomatal model performed, several model statistics were*
*calculated. These model statistics are explained below.*

The mean bias (MB) is the mean difference between the simulations ($S_i$) and the observations ($O_i$),
with n being the number of data points (Stone, 1993);

$$MB = n^{-1} \sum_{i=1}^{n} (S_i - O_i) \tag{B1}$$

The mean relative error (MRE) is the mean relative difference between the simulations and the
observations (Peierls, 1935);

$$MRE = n^{-1} \sum_{i=1}^{n} \frac{|S_i - O_i|}{O_i} \tag{B2}$$

Willmott's index of agreement (d) is a dimensionless goodness-of-fit coefficient, with $\bar{O}$ being the
mean observation (Willmott, 1981); The index can vary between 0 and 1, with d equals 1 for a perfect
agreement between simulations and observations.

$$d = 1 - \frac{\sum_{i=1}^{n} (S_i - O_i)^2}{\sum_{i=1}^{n} (|S_i - \bar{O}| + |O_i - \bar{O}|)} \tag{B3}$$

The model efficiency (ME) gives an indication of how well the observations match the simulations
(Nash & Sutcliffe, 1970); Model efficiency can range from -∞ to 1 and is 1 when simulations and
observations match perfectly. An efficiency of 0 indicates that the simulations are as accurate as the
mean observation and an efficiency of less than zero indicates that the mean observation is a better
predictor than the model.

$$ME = 1 - \frac{\sum_{i=1}^{n} (S_i - O_i)^2}{\sum_{i=1}^{n} (O_i - \bar{O})^2} \tag{B4}$$

The root-mean-squared error (RMSE) is a measure of the mean absolute difference between the
simulations and the observations, weighting large differences heavily (Willmott et al., 1985); The
systematic component ($RMSE_s$) estimates the model's linear or systematic error, hence, the better the
regression between simulations and observations, the smaller the systematic component (Willmott et
al., 1985). The unsystematic component is a measure of how much of the discrepancy between
simulations and observations is due to random processes (Willmott et al., 1985). A good model will
provide low values of RMSE, with $RMSE_s$ close to zero and $RMSE_u$ close to RMSE (Willmott et al.,
1985).

$$RMSE = \sqrt{n^{-1} \sum_{i=1}^{n} (S_i - O_i)^2} \tag{B5}$$





$$RMSE_s = \sqrt{n^{-1} \sum_{i=1}^{n} (S'_i - O_i)^2}$$ (B6)

$$RMSE_u = \sqrt{n^{-1} \sum_{i=1}^{n} (S_i - S'_i)^2}$$ (B7)

$S'_i = a * O_i + b$ , where 'a' and 'b' are slope and intercept, respectively, of the linear regression of the simulations versus the observations.





**Appendix C The canopy model**

*Stomatal conductance was calculated on leaf level with the stomatal model. The canopy model was used to scale up these values for the whole canopy. Ozone fluxes were calculated, based on an electrical conductance analogy. Below the general canopy model, including conductance analogy model, is*
*explained, followed by the explanation of two submodel that were used: solar elevation submodel and radiation submodel.*

**Part 1 The canopy model**

The canopy model is an algorithm to scale up $g_{st}$ at leaf level to $g_{st}$ at canopy level. Subsequently, $O_3$ fluxes are calculated with an electrical conductance analogy model, which calculates the instant canopy
$O_3$ uptake from different input data Rg, $T_{air}$, VPD, LAI, SWP, and $O_3$ concentration. The canopy model consists of three submodels: the multiplicative stomatal model (Appendix A), the solar elevation submodel, and the radiation transfer submodel (see below).

First, the canopy is divided into different horizontal layers, with each a sunlit and shaded fraction. For each layer fraction the incoming PAR is calculated with the radiation transfer submodel. With the
multiplicative stomatal model $g_{st}$ is calculated for each layer fraction.

For each layer fraction the total leaf conductance (mol m$^{-2}$ leaf area s$^{-1}$) is calculated by summing $g_{st}$ and $g_{ns}$, the non-stomatal conductance. This value is multiplied by LAI of the layer fraction and the values for both the sunlit and the shaded layer fraction are summed to obtain the total layer conductance (mol m$^{-2}$ ground area s$^{-1}$). All layer conductances can be summed to obtain the conductance of the canopy as
a whole ($g_{can}$).

The total conductance is a function of $g_{bl}$ and $g_{can}$ based on an electrical conductance analogy model.

$$g_{tot} = (\frac{1}{g_{aero}} + \frac{1}{g_{bl}} + \frac{1}{g_{can}})^{-1} \tag{C1}$$

where $g_{aero}$ is the aerodynamic conductance; $g_{bl}$ is the boundary layer conductance; $g_{can}$ is the canopy conductance.

Total $O_3$ flux (nmol m$^{-2}$ ground area s$^{-1}$) is the $O_3$ flux for the whole canopy and is then calculated by:

$$F_{tot} = [O_3] * g_{tot} \tag{C2}$$

where $[O_3]$ is the $O_3$ concentration (ppb).

Stomatal $O_3$ flux is the fraction of the total $O_3$ flux taken up by the stomata and is calculated by:

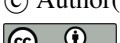



$$F_{st} = [O_3] * \frac{g_{st}}{g_{st} + g_{ns}} \tag{C3}$$

where $g_{st}$ is the stomatal conductance (mol m$^{-2}$ ground area s$^{-1}$); $g_{ns}$ is the non-stomatal conductance (mol m$^{-2}$ ground area s$^{-1}$).

Non-stomatal O$_3$ flux (F$_{ns}$) is the difference between total O$_3$ flux and stomatal O$_3$ flux:

$$F_{ns} = F_{tot} - F_{st} \tag{C4}$$

**Part 2 The solar elevation submodel**

This submodel calculates the solar elevation angle, β (radians), at each time step (Campbell and Norman, 1998).

$$β = \arcsin (\sin \phi \sin \delta + \cos \phi \cos \delta \cos h) \tag{C5}$$

where $\delta$ is the solar declination angle;           $\delta = -23.4 * \left(\frac{\pi}{180}\right) * \cos(2 * \pi * \frac{doy+10}{365})$

$\phi$ is the latitude in radians;           $\phi = 0.89$

$h$ is the hour angle of the sun;           $h = \pi * \frac{t - t_0}{12.0}$

where t is time; $t_0$ is solar noon;           $t_0 = 12 + \frac{4*(L_s - L_e) - E_t}{60.0}$

$L_s$ is the standard longitude in degrees;    $L_s = 15.0$

$L_i$ is the local longitude in degrees;       $L_i = 4.0$

$E_t$ is the empheris of the sun;

$E_t = 0.017 + 0.4281 * \cos(F_d) - 7.351 * \sin(F_d) - 3.349 * \cos(2 * F_d) - 9.731 * \sin(F_d)$

where $F_d$ is the day angle;           $F_d = 2 * \pi * \frac{doy-1}{365}$

**Part 3 The radiation transfer submodel**

The radiation submodel calculates the direct (I$_{b0}$) and diffuse (I$_{d0}$) fraction of the incoming radiation (I) at the top of the canopy. Hence, I is equal to R$_g$. These calculation is based on the difference between




measured and theoretically potential incoming radiation above the canopy, which is depending on β, the
solar elevation angle (Op de Beeck et al., 2010).

First the sunlit LAI fraction of each horizontal leaf layer i is calculated with Beer's law:

$$f_{sun(i)} = \exp(-k_b \Omega LAI_{c(i)}) \tag{C6}$$

where $k_b$ is the direct radiation extinction coefficient; $\Omega$ is a factor accounting for inter- and intra-crown
foliage clumping; $LAI_{c(i)}$ is the cumulative LAI above a horizontal leaf layer i.

A spherical needle angle distribution is assumed, hence $k_b = 0.5/\sin\beta$ (de Pury, 1997).

The shaded LAI fraction of each horizontal leaf layer i is calculated as follows:

$$f_{shad(i)} = 1 - f_{sun(i)} \tag{C7}$$

where fsun(i) is the sunlit LAI fraction.

The intensity of direct radiation does not decline through the canopy, but the diffuse radiation does and
is calculated with Beer's law:

$$I_{d(i)} = I_{d0} * \exp(k_d \Omega LAI_{c(i)}) \tag{C8}$$

where $I_{d0}$ is the incoming diffuse radiation.

The total received radiation by a sunlit fraction ($I_{sun(i)}$) is the sum of direct and diffuse radiation. Shaded
leaves only receive diffuse radiation:

$$I_{shad(i)} = I_{d(i)} \tag{C9}$$

$$I_{sun(i)} = \cos\left(\frac{\pi}{3}\right) * I_{b0} + I_{d(i)} \tag{C10}$$

where $\left(\frac{\pi}{3}\right)$ is the averaged leaf angle for a uniform needle angle distribution; $I_{b0}$ is the incoming direct
radiation at top of the canopy; $I_{d(i)}$ is the diffuse radiation for a horizontal leaf layer i.

Total received irradiance is now converted to total received PAR and split into PAR per horizontal leaf
layer.





**Author contribution** L Verryckt, M. Op de Beeck, B. Gielen, M. Roland and I.A. Janssens designed the study.
J. Neirynck provided the $O_3$ concentration measurements, B. Gielen provided the EC and LAI data, B. Gielen,
M. Op de Beeck and L. Verryckt measured $g_{st}$ in sity, and M. Op de Beeck and L. Verryckt conducted the
modelling. All authors contributed to the writing.

**Acknowledgement** The measurements for this work were funded by the Hercules Foundation, through support
of the Brasschaat ICOS ecosystem station. IAJ acknowledges support from the European Research Council
Synergy grant ERC-2013-SyG-610028 IMBALANCE-P.

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





Table 1. Optimised parameter values of the multiplicative stomatal model.

| | |
|---|---|
| $g_{min}$ (mol $O_3$ $m^{-2}$ $s^{-1}$) | 0.02 |
| $g_{max}$ (mol $O_3$ $m^{-2}$ $s^{-1}$) | 0.14 |
| a | 0.0057 |
| $T_{opt}$ (°C) | 25.61 |
| $T_{min}$ (°C) | 5.47 |
| $VPD_{min}$ (kPa) | 3.16 |
| $VPD_{max}$ (kPa) | 0.51 |
| $SWP_{min}$ (MPa) | -1.18 |
| $SWP_{max}$ (MPa) | -0.19 |



Table 2. Statistics of the model evaluation. The statistics used to evaluate the model performance are mean bias (MB), relative mean error (RME), systematic and unsystematic root mean squared error ($RMSE_{s/u}$), Willmott's

index of agreement (d), model efficiency (ME), coefficient of determination ($R^2$).

| Statistics | Parameterisation | Validation |
|---|---|---|
| MB | 0.002 | 0.002 |
| RME | 0.34 | 0.33 |
| RMSE | 0.019 | 0.019 |
| $RMSE_s$ | 0.006 | 0.006 |
| $RMSE_u$ | 0.017 | 0.017 |
| d | 0.99 | 0.99 |
| ME | 0.72 | 0.72 |
| $R^2$ | 0.72 | 0.72 |

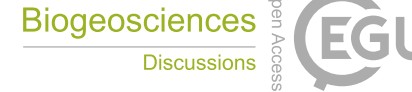



Table 3. GPP values (mol C m$^{-2}$ day$^{-1}$) for the days with low stomatal O$_3$ uptake, high stomatal O$_3$ uptake, and the growing season total (GS) of different years as well as the multi-annual mean. One GPP model (GPPmod1) was trained without days with high stomatal O$_3$ uptake, whereas a second GPP model (GPPmod2) also excluded days following high O$_3$ events, and was thus trained to test for lag effects of O$_3$. The relative difference between modelled and EC-derived GPP estimates is presented between brackets. Positive values indicate an overestimation by the model and therefore a potential O$_3$ effects on GPP.

| Year | Days with low stomatal O$_3$ uptake | | | Days with high stomatal O$_3$ uptake | | | Growing Season Total | | |
|---|---|---|---|---|---|---|---|---|---|
| | GPPmea | GPPmod1 | GPPmod2 | GPPmea | GPPmod1 | GPPmod2 | GPPmea | GPPmod1 | GPPmod2 |
| 1998 | 0.47 | 0.43 (-7%) | 0.38 (-20%) | 0.76 | 0.56 (-2%) | 0.57 (-26%) | 0.48 | 0.44 (-9%) | 0.39 (-22%) |
| 2000 | 0.48 | 0.46 (-4%) | 0.48 (+0%) | 0.70 | 0.65 (-7%) | 0.70 (-0%) | 0.49 | 0.47 (-4%) | 0.49 (+0%) |
| 2001 | 0.43 | 0.42 (-1%) | 0.37 (-13%) | 0.64 | 0.51 (-20%) | 0.45 (-29%) | 0.44 | 0.43 (-3%) | 0.38 (-14%) |
| 2002 | 0.39 | 0.40 (+4%) | 0.38 (-2%) | 0.57 | 0.52 (-7%) | 0.50 (-11%) | 0.40 | 0.41 (+3%) | 0.39 (-2%) |
| 2004 | 0.42 | 0.40 (-3%) | 0.41 (-2%) | 0.56 | 0.55 (-3%) | 0.59 (+4%) | 0.43 | 0.41 (-3%) | 0.42 (-2%) |
| 2005 | 0.41 | 0.40 (-3%) | 0.41 (-1%) | 0.62 | 0.57 (-7%) | 0.56 (-9%) | 0.42 | 0.41 (-3%) | 0.42 (-1%) |
| 2006 | 0.45 | 0.41 (-10%) | 0.37 (-19%) | 0.76 | 0.58 (-24%) | 0.45 (-42%) | 0.47 | 0.42 (-13%) | 0.37 (-24%) |
| 2007 | 0.44 | 0.45 (+2%) | 0.45 (+3%) | 0.76 | 0.64 (-17%) | 0.64 (-17%) | 0.46 | 0.46 (-2%) | 0.46 (-1%) |
| 2008 | 0.47 | 0.47 (-0.4%) | 0.67 (+44%) | 0.84 | 0.75 (-10%) | 0.74 (-12%) | 0.49 | 0.48 (-2%) | 0.68 (+40%) |
| 2009 | 0.51 | 0.49 (-4%) | 0.64 (+24%) | 0.80 | 0.78 (-3%) | 0.76 (-6%) | 0.57 | 0.51 (-12%) | 0.64 (15%) |
| 2010 | 0.47 | 0.53 (+13%) | 0.57 (+21%) | 0.74 | 0.74 (+0%) | 0.76 (+3%) | 0.49 | 0.54 (+10%) | 0.58 (+16%) |
| 2011 | 0.60 | 0.55 (-8%) | 0.55 (-8%) | 0.82 | 0.76 (-7%) | 0.80 (-2%) | 0.64 | 0.56 (-14%) | 0.56 (-14%) |
| 2012 | 0.64 | 0.62 (-3%) | 0.63 (-2%) | 1.06 | 0.92 (-13%) | 1.01 (-5%) | 0.66 | 0.64 (-4%) | 0.64 (-3%) |
| 2013 | 0.52 | 0.56 (+7%) | 0.56 (+6%) | 0.81 | 0.79 (-2%) | 0.92 (+14%) | 0.55 | 0.58 (+5%) | 0.58 (+5%) |
| Mean | 0.48 | 0.47 | 0.49 | 0.75 | 0.67 | 0.67 | 0.50 | 0.48 | 0.50 |





Fig. 1. Fingerprint of the meteorological and eddy-flux derived gross primary productivity (GPP) measurements averaged over the period 1998-2013. Day of year is plotted on the y-axis and hour of day on the x-axis. Air temperature ($T_{air}$), incoming global radiation (Rg), vapour pressure deficit (VPD), and GPP are plotted.

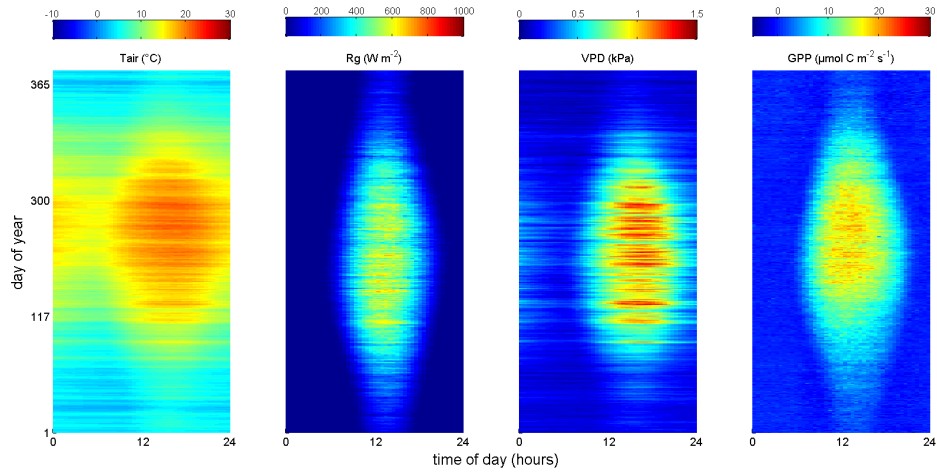






Fig. 2. Time series of the weekly total precipitation and mean soil water potential (SWP). The precipitation and SWP data are averaged over the period 1998-2013. Error bars represent the 95% confidence intervals.

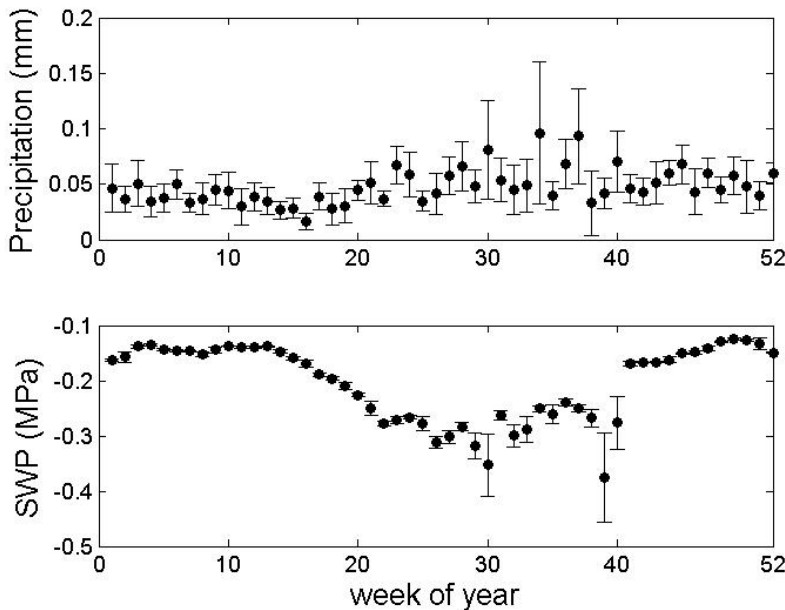




Fig. 3. Seasonal changes of LAI over the 15 year study period at the Brasschaat Scots pine site.

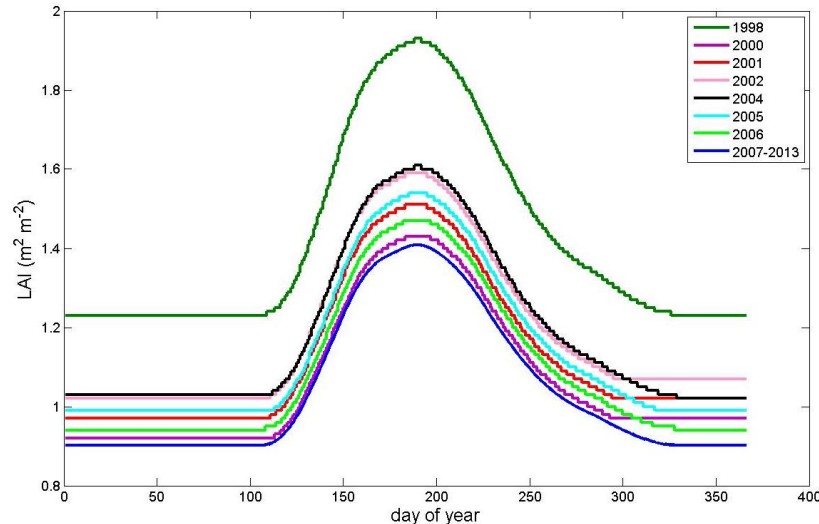






Fig. 4. Measured versus modelled stomatal conductance ($g_{st}$) for parameterisation dataset (A) (n = 205) and validation dataset (B) (n = 205). The black line is the 1:1 line. The red line is the linear fit for which the equation is given in the figure. The p-values of the slope ($p_a$) and the intercept ($p_b$) are also shown.

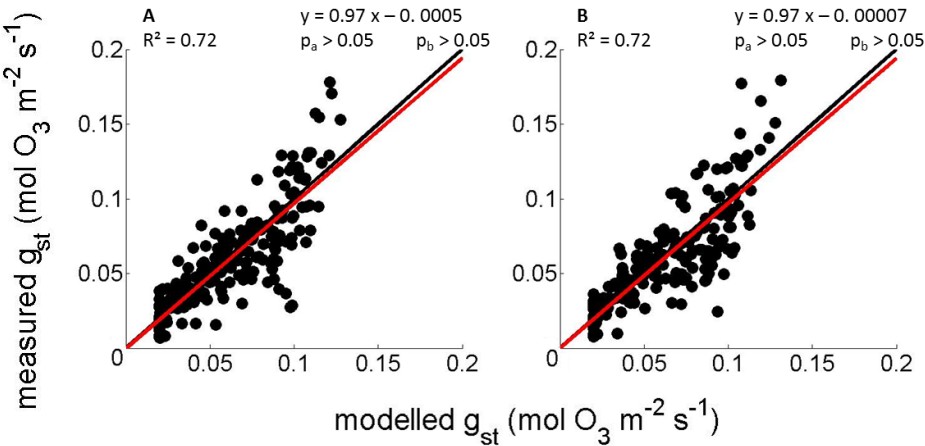






Fig. 5. Measured stomatal conductance ($g_{st}$) in function of the different variables used in the multiplicative model: photosynthetically active radiation (PAR), air temperature ($T_{air}$), vapour pressure deficit (VPD), and soil water potential (SWP). The red line represents the boundary line for which the functions are given in Appendix A (A3-

A6). (n = 205)

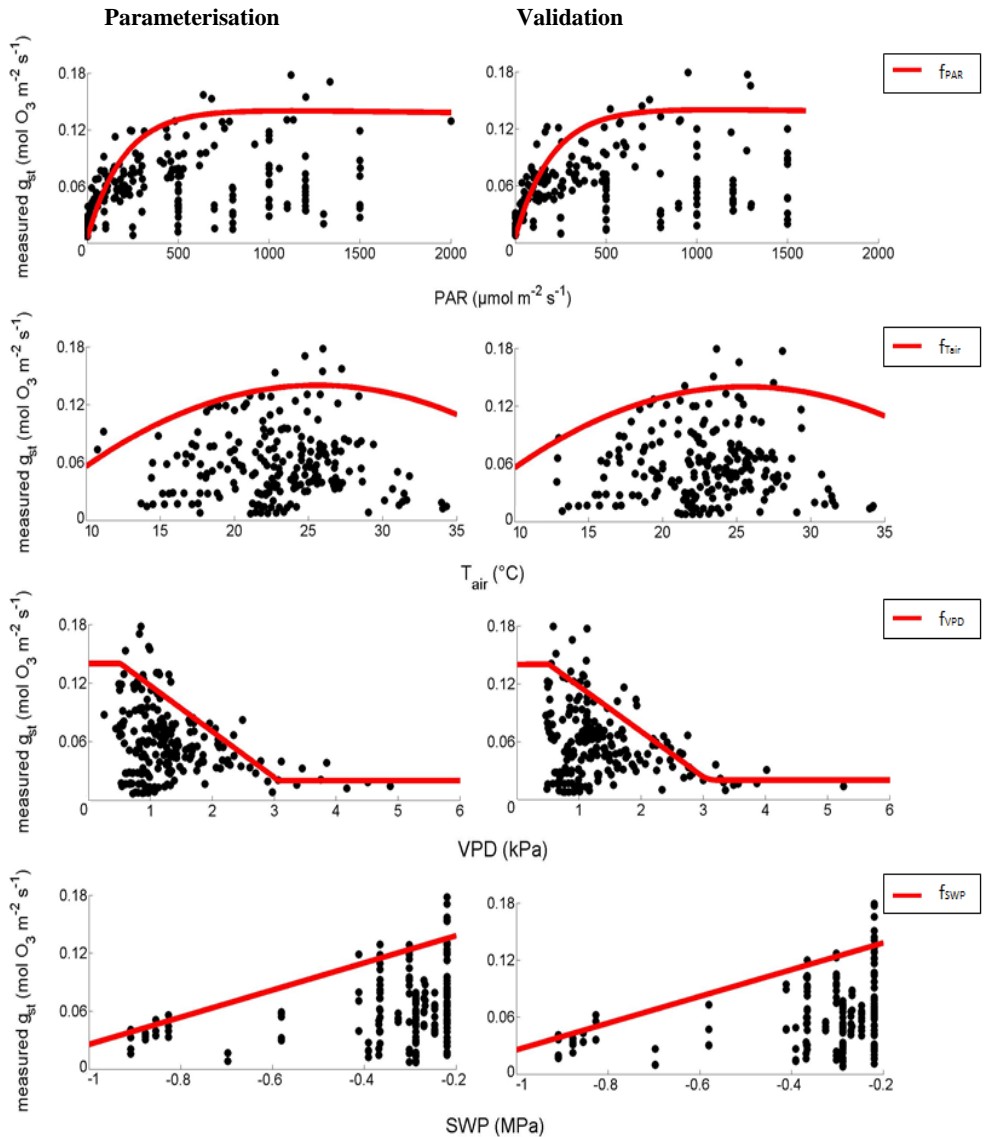





Fig. 6. Fingerprints of stomatal conductance ($g_{st}$), ozone concentration ($[O_3]$), and stomatal ($F_{st}$), non-stomatal ($F_{ns}$), and total ozone flux ($F_{tot}$). Day of year is plotted on the y-axis and hour of day on the x-axis. Please note the different scales for $F_{st}$, $F_{ns}$, and $F_{tot}$.

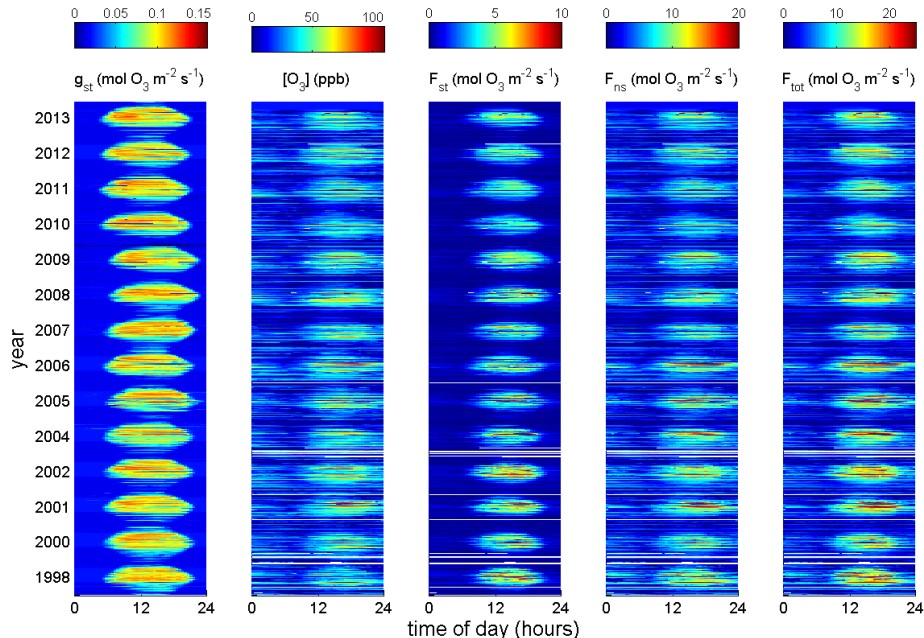





Fig. 7. Gross primary productivity (GPP)-model residuals in function of total stomatal ozone flux over the growing season ($F_{st}$; panels A, B), AOT40 (panels C, D), $POD_1$ (panels E, F), and $POD_2$ (panels G, H). The GPP model of (A), (C), (E), and (G) was trained without days with high stomatal $O_3$ uptake, whereas the GPP model of (B), (D), (F), and (H) was trained to test for possible lag effects of $O_3$ on GPP. The vertical dashed lines represent the threshold values used in Europe (C – H). For each relation a linear function was fitted, but none were statistically

significant as indicated in the panels (n = 14).





