# Peer review of "Fig. S1. Fingerprint of annual anomalies of the meteorological, soil water, and carbon exchange measurements compared to the averaged values shown in Fig. 1 and 2 of the main text. For each day of the year air temperature (Tair), incoming global radiation (Rg), vapour pressure deficit (VPD), soil wa"

_Biogeosciences, 2016_

## Referee Comment (RC1) · Anonymous Referee #3 · 23 Mar 2016

The paper aims to assess O3 damage on vegetation in a Belgian pine forest. The adopted methodology is based on the assumption that, if an O3 detrimental effect occurs, a GPP model parameterized for days with low O3 stomatal uptake would overestimate GPP during days with high O3 stomatal uptake. In my opinion, this assumption is not valid. As the authors themselves suggested (lines 267-272), solar radiation is usually high during high O3 events, so that a model parameterized under low O3 condition would be also parameterized under low irradiation, leading to a GPP underestimation. The authors poorly discussed the confounding effect of solar radiation on the study. In this frame, it would be desirable to add more information about ANN processes and point out the importance of solar radiation on the ANN operation (i.e. sensitivity analysis). The paper is well structured and clearly written, it could be a novel work about O3 detrimental effect on vegetation if some changes in the method were made. I would
encourage this manuscript to be resubmitted after taken into account the confounding effect of solar radiation on the results.

A few specific comments are suggested below: Line 192: Why is g_areo set to 1? Lines 213-218: Authors should provide more information about network's building. Please clarify: How did the authors choose the nodes number? Before training the network , did the authors scaled inputs and target data so that their magnitudes were similar (e.i. [0 1] or [-1 1])?

Some technical corrections: Line 25: Do authors mean "IPCC, 2007" instead "IPCC, 2001"? Otherwise, insert citation in the reference list. Line 29: Use "O3" instead of "ozone" to be consistent along the text. Lines 30-31, 295, 350: Insert spaces between semicolons and references' names. Line 33: Do authors mean "affect" instead of "effect"? Lines 88-112: Be consistent with verb tenses, use past forms. Line 296: "the empirical the dose-response", replace with "the empirical dose-response". Line 344: Figure 6 is too condensate, I cannot notice what authors write: "This difference between AOT40 and POD1 during 2006 was due to stomatal closure; during high O3 concentration events, gst was rather low (Fig. 6)". I suggest to eliminate Fst Fns and Ftot plots from the figure and to show gsto and [O3] horizontally. Please, inset letters to each plot of the figures. Line 520: replace "in sity" with "in situ".

---

## Referee Comment (RC2) · Anonymous Referee #4 · 26 May 2016

This paper is very interesting because the studies on the response of mature trees to ozone are not so many. I strongly encourage authors to submit it again in the light of remarks

My major criticism of this work is as follows:

The basis of the authors' reasoning is that the effects of ozone on the functioning of trees are fleeting, and last only a few days. It is unfortunate that the authors do not give references confirming this statement, because I think that it is not obvious. The literature suggests that the effects of ozone may extend over the long term, particularly by activation of defense gene or induction of senescence, especially when ozone levels are relatively high. It seems to be the case in this study, since the authors indicate that the usual critical levels (AOT40 and PODy) are exceeded each year.

[Figure]

Therefore, if the trees have already experienced "high ozone fluxes days" before the days of "low ozone fluxes" selected for calibrating the model, it becomes difficult to argue that the behavior of the trees in "low ozone fluxes days" is the same as if they had never been exposed to the pollutant. On the contrary, it makes sense that there is little difference between the two situations.

I wish that the authors give more convincing arguments on this point, and justify the assumption that the trees behave in the "low ozone flux" days in the same way as if they had never been exposed to ozone.

- The question of the validity of the use of a model calibrated for low-radiation conditions to calculate fluxes in high radiation conditions should also be more justified

Nevertheless this work is interesting, and the paper is well written.

I have still some remarks, which are just details:

- The 0.61 conversion factor between the conductance values for water and ozone is questionable, many authors use 0.663 instead (see eg Grünhage et al, 2013). However, this did not have much impact on the results presented.

- The specificity of this cover (sparse canopy), and the validity of fluxes measurements on these types of vegetation could be discussed further.

- Many cited references are quite old (half are over 10 years). Are there not more recent references?
* * *

---

## Author Comment (AC1) · 23 Jun 2016

Dear referee

Thank you for your comments on this manuscript and the thoughtful suggestions to increase its scientific quality. Below you find our response to each of your main comments.

The referee's comment: 1. The adopted methodology is based on the assumption that, if an O3 detrimental effect occurs, a GPP model parameterized for days with low O3 stomatal uptake would overestimate GPP during days with high O3 stomatal uptake. In my opinion, this assumption is not valid. As the authors themselves suggested (lines 267-272), solar radiation is usually high during high O3 events, so that a model parameterized under low O3 condition would be also parameterized under low irradiation, leading to a GPP underestimation. The authors poorly discussed the confounding effect of solar radiation on the study.

The authors' answer: We understand this comment since we didn't provide much details about the radiation conditions on the days we used to train the ANN model and on the days with high O3 uptake, on which we expected an O3 effect. It was however not the case that low and high solar radiation conditions were both exclusive to one of the datasets. (Our statement on lines 267-272 about solar radiation being usually high during O3 events is in this regard misleading). This is demonstrated in the upper left graph of Fig. 1 below, which shows the histograms of daily total radiation (Rg) for the training dataset (red) and the high O3 uptake dataset (blue). While it is true that Rg is on average rather high on days with high O3 uptake, as we suggest on lines 267-272, the figure shows that the training dataset included equally well days with high Rg as days with low Rg. Moreover, the training dataset covered an Rg range that contains the Rg range of the high O3 uptake dataset. We hence believe the model is unlikely to include a bias that is due to poor parameterization at high Rg and that leads to GPP underestimation for days with high O3 uptake. We have plotted in the figure below also the histograms for temperature and VPD, two other main drivers of GPP. Also for these two variables the training dataset covers a variable range that contains the variable range of the high O3 uptake dataset. In general, we believe that the training dataset and the high O3 uptake dataset overlap sufficiently in the 'multi parameter space' for the model not to include a bias that would result in a GPP overestimation that we might wrongly interpret as an effect of O3. This information and the graphs below will be added to the revised manuscript to inform all readers.

The referee's comment: 2. Line 192: Why is g_areo set to 1?

The authors' answer: The aerodynamic conductance was set to a fixed value of 1, because we were not able to calculate g_aero accurately from other measured parameters and because g_aero was expected to be very high relative to the other conductances, as O3 concentrations were measured at only 3 meters above the (rather

smooth) canopy surface. For these two reasons we set g_aero to a rather high and fixed value of 1 nmol m-2 s-1. We admit that the choice for this value lacks further scientific basis. Therefore, in the new analyses for the revised manuscript, we will put this value infinitely high. In other words, we will omit g_aero from the conductance scheme. This will only have a negligible effect on the calculated O3 fluxes.

The referee's comment: 3. Lines 213-218: Authors should provide more information about network's building. Please clarify: How did the authors choose the nodes number? Before training the network, did the authors scaled inputs and target data so that their magnitudes were similar (e.i. [0 1] or [-1 1])?

The authors' answer: The numbers of layers and nodes were chosen after testing the network with different numbers of layers and/or nodes. Since the model is not too complex, the model does not need many layers and nodes. The model with 1 layer and 10 nodes gave the best $R^2$ value for the validation dataset for which also a high $R^2$ value was given for the training dataset (see Fig. 2). A normalization process was applied for training and testing the data. The default settings of the Matlab Neural Network Toolbox were used. Data were scaled to [-1 1] based on the lowest and highest value in the dataset. We will add a line to the manuscript to inform the reader that we tested different models and the network with 1 layers of 10 nodes came out as the best performing. We wouldn't show Fig. 2 below with all results, though. We will also add the information about how data were scaled to [-1 1] in the Matlab Toolbox.

The referee's comment: 4. Some technical corrections.

The authors' answer: We agree with all technical corrections and will adjust these in the revised manuscript. We would like to thank you again for the helpful comments and suggestions on the manuscript.
* * *
[Figure]

[Figure]

**Fig. 1.** Histograms of meteorological variables for the trainingsdataset (red) and the high O3 uptake dataset (blue).

| Layers | Nodes | R² - training | R² - validation |
|--------|-------|---------------|-----------------|
| 1 | 1 | 0,6064 | 0,6239 |
| 1 | 5 | 0,7881 | 0,7293 |
| 1 | 10 | 0,8457 | 0,7656 |
| 1 | 15 | 0,8647 | 0,7534 |
| 1 | 20 | 0,9603 | 0,6959 |
| 1 | 50 | 0,9632 | 0,6055 |
| 2 | 1 | 0,6136 | 0,6088 |
| 2 | 5 | 0,7967 | 0,7711 |
| 2 | 15 | 0,9469 | 0,7367 |
| 2 | 20 | 0,8808 | 0,7146 |
| 10 | 1 | 0,621 | 0,5959 |
| 10 | 10 | 0,8924 | 0,7476 |
| 10 | 50 | 0,9969 | 0,4989 |

**Fig. 2.** Layers and nodes tested in a neural network with the $R^2$ value of the training and validation dataset.

---

## Author Comment (AC2) · 23 Jun 2016

Dear referee

We thank you for your constructive criticism, and the time invested in the analysis of this manuscript. Our responses and explanations related to your comments are listed below.

The referee's comment: 1. The basis of the authors' reasoning is that the effects of ozone on the functioning of trees are fleeting, and last only a few days. It is unfortunate that the authors do not give references confirming this statement, because I think that it is not obvious. The literature suggests that the effects of ozone may extend over the long term, particularly by activation of defense gene or induction of senescence, especially when ozone levels are relatively high. It seems to be the case in this study,

since the authors indicate that the usual critical levels (AOT40 and PODy) are exceeded each year. Therefore, if the trees have already experienced "high ozone fluxes days" before the days of "low ozone fluxes" selected for calibrating the model, it becomes difficult to argue that the behavior of the trees in "low ozone fluxes days" is the same as if they had never been exposed to the pollutant. On the contrary, it makes sense that there is little difference between the two situations. I wish that the authors give more convincing arguments on this point, and justify the assumption that the trees behave in the "low ozone flux" days in the same way as if they had never been exposed to ozone.

The authors' response: The reasoning behind our analysis is indeed that O3 effects on gross C uptake only last on the short term. (We assume that they occur through e.g. damage to the photosynthetic apparatus after O3 uptake peaks, when the detoxification capacity of the tree is exceeded, and that this damage is repaired after the peak, when O3 loads have fallen back to low levels.) Nevertheless, we agree with you that a carry-over effect cannot be ruled out and that we need to somehow demonstrate that the trees exposed to low O3 fluxes late in the growing season behave in the same way as when exposed to similar low O3 fluxes early in the growing season. To demonstrate this, we compiled a dataset that contained per growing season only the days after the first major O3 flux peak in the growing season. From these days, we further selected only the ones with low O3 fluxes (we excluded the O3 flux peaks) and no short-term effect expected (we excluded also the first six days after the O3 flux peaks). We trained the GPP model with these data and then predicted GPP for the days occurring before the first major O3 peak in each growing season. If a carry-over effect would be present, it would be somehow included in the trained model. This would then underestimate GPP for the days before each first major O3 peak, where a carry-over effect is assumed not yet to have occurred. We evaluated and compared the linear regressions of measured versus modelled GPP (GPPmea and GPPmod) of both datasets (Figure 1). For both regressions, intercept and slope were not significantly different from 0 and 1 respectively (training: pslope = 1, pintercept = 1, testing: pslope = 0.83, pintercept = 0.44). The slopes were also not significantly different from each other (p = 0.86)

and neither were the intercepts (p = 0.53). From these results, we infer the model did not underestimated GPP for the days before each first major O3 peak in the season. Based on this analysis, we conclude that carry-over effects of O3 were unlikely to have occurred, since not detectable with this approach at a statistically significant level, and that our assumption on the absence of (detectable) carry-over effects is valid. We will add this information to the revised manuscript.

Note: Testing for a carry-over effect by training the model with the days before each first major O3 flux peak and then evaluating GPP predictions for low O3 days after each first major O3 peak was not possible, because there were too few data for good model training.

An additional note: Exceedance of the critical levels for AOT40 and POD1 does not necessarily imply long term O3 damage in the Scots pine forest in our study. These critical levels are determined based on experiments on seedlings (Karlsson et al., 2004) and the critical level of POD1 for Scots pine was adopted from the critical level for Norway spruce since no data on Scots pine is available (Mills 2011). The critical levels are, in general, based on the best available knowledge, so they are not definitive or absolute (Skärby et al., 1998).

The referee's comment: 3. The question of the validity of the use of a model calibrated for low-radiation conditions to calculate fluxes in high radiation conditions should also be more justified.

The authors' response: Here, we would like to refer to our response letter to referee #1, who raised the same comment. We show with histograms that the parameterisation dataset includes equally well days with high radiation (Rg) and days with low Rg. Moreover, the Rg range of the parameterization set fully contains the Rg range in the high O3 uptake data. We are confident that (1) our model does not include a bias due to underrepresentation of days with high Rg in the dataset used for calibration and (2) that this dataset and the high O3 update dataset overlay sufficiently in the

multi-parameter space, including Rg. Therefore our GPP model is not only calibrated for low radiation conditions, and as the other histograms included in the response letter demonstrate, neither for only low temperature or VPD. We will add this information to the revised manuscript.

The referee's comment: 4. I have still some remarks, which are just details:

- The 0.61 conversion factor between the conductance values for water and ozone is questionable, many authors use 0.663 instead (see eg Grünhage et al, 2013). However, this did not have much impact on the results presented.

The authors' response: The value we used in this paper is based on the Graham's law of diffusion (Mason and Kronstadt 1967). We agree that sometimes a slightly higher conversion factor is applied. But as you suggest, using a value of 0.663 instead of 0.61 will not have a significant impact on our results. Therefore we decide not to change this value in our analyses.

- The specificity of this cover (sparse canopy), and the validity of fluxes measurements on these types of vegetation could be discussed further.

The authors' response: We do not fully understand what the referee is pointing to with this comment. We do not think additional information about the specificity of the cover and the validity of fluxes measurements on pine forests will be relevant to this manuscript.

- Many cited references are quite old (half are over 10 years). Are there not more recent references?

The authors' response: We will take this into account when revising the manuscript. Where deemed useful, we will add more recent references or replace older with more recent references.

Caption of Fig. 1: GPPmea as function of GPPmod for two datasets: (a) only the days before the first major O3 peak in every year, (b) the training dataset with the days after

the first major O3 peak in every year, excluding those with high O3 fluxes + six following days. The black line is the 1:1 line. The blue line is the regression fit including 95% confidence intervals (in grey).

―――――――――――――――――――

[Figure]

Fig. 1. GPPmea as function of GPPmod for two datasets: (a) only the days before the first major O3 peak in every year, (b) the training dataset with the days after the first major O3 peak in every year.

---

## Author Response (AR1)

**Dear Editor,**

I am pleased to resubmit the revised version of our manuscript "No impact of tropospheric ozone on the gross primary productivity of a Belgian pine forest". We appreciated the referees' constructive criticisms, and yours, as associate editor. I have addressed each of the concerns as outlined below.

**Associate editor comments:**

- 1. In general,  $O_3$  with the unit ppb should be referred to as mixing ratio instead of concentration. Thank you for this remark. We changed this throughout the manuscript.
- 2. It is unclear to the referee, why the aerodynamic conductance could not be calculated from the data retrieved by the eddy covariance measurements (e.g., u\*)? I think a multi-layer approach that considers all conductances of the canopy layers (also the in-canopy aerodynamic conductance) would be more appropriate than a big-leaf approach, but therefore a measured or known vertical profile of the O3 mixing ratio (and wind speed) has to be known.

We calculated the aerodynamic conductance based on the friction velocity  $u^*$  and the atmospheric stability function  $\Psi$ h using the set of coefficients published by Dyer (1974).

3. Appendix C, equation C3: this equation appears to be wrong. The mixing ratio of O3 is multiplied by a dimensionless factor given by a ratio of conductances. The resulting unit would be a mixing ratio and not a flux density with the unit nmol  $m^{-2} s^{-1}$ . The conventional equation to retrieve  $F_{st}$  is (e.g. Gerosa et al., 2005):

Fst = gst \* O3 (canopy)

Here,  $O_3$  (canopy) is the  $O_3$  mixing ratio just above the canopy top or at the zero plane displacement height. Note, that this mixing ratio should not be identical to the one used in equation C2 (which is the  $O_3$  mixing ratio at the measurement height). Obviously, the overall results of the manuscript have to be revised if equation C3 was indeed used to derive Fst. The conclusions may change accordingly.

This was a textual error and is corrected in the revised manuscript. Analyses did not need to be revised because of this.

4. Technical corrections

We agreed with all of them and changed them accordingly.

**Referee #3 comments:**

1. Clarification of arguments for using of a GPP model parameterized for days with low O3 stomatal uptake including detailed discussion/rebuttal.

We added information about this comment as suggested in the answer to this referee. This can be found at line 242-248 in 'Materials and Methods – 2.5 Detecting  $O_3$  effects on GPP', at line 293-298 in 'Results – 3.3 Ozone effects on GPP' and at line 361-365 in 'Discussion – 4.3 Ozone effects on GPP'.

**Referee #4 comments:**

1. Justification of assumption that trees behave the same under "low ozone flux" days as if they were never exposed to ozone

Information about this assumption was added to the manuscript as suggested in the answer to this referee: in 'Materials and Methods – 2.5 Detecting  $O_3$  effects on GPP' from line 249-26,

in 'Results – 3.3 Ozone effects on GPP' from line 299-303 and in 'Discussion – 4.3 Ozone effects on GPP' from line 365-368.

**Additional changes:**

In addition to the comments, we applied some changes throughout the document to improve the readability. This resulted in a subdivision of Materials and Methods, mainly in the 'Measurements' section. We applied more statistical analyses to endorse our results. This information can be found under 'Materials and Methods - 2.5 Detecting O3 effects on GPP', resulting in additional information in 'Results – 3.3 Ozone effects on GPP' (line 304-318) and in the 'Discussion – 4.3 Ozone effects on GPP' (line 369-374).

We removed table 3 from the manuscript and replaced it by figure 8 which more clearly represents the results. We also reviewed figure 9.

**No impact of tropospheric ozone impact on the gross primary uptakecarbon uptake ofby a Belgian pine forest**

Lore T. Verryckt1, Maarten Op de Beeck1, Johan Neirynck2, Bert Gielen1, Marilyn Roland1, and Ivan A. Janssens1

1Department of Biology, University of Antwerp, Wilrijk, 2610, Belgium
 2Research Institute for Nature and Forest, Geraardsbergen, 9500, Belgium
 *Correspondence to:* L. Verryckt (lore.verryckt@uantwerpen.be)

Abstract High stomatal ozone  $(O_3)$  uptake has been shown to negatively affect crop yields and the growth of tree 10 seedlings. However, little is known about the effect of O3 on the carbon uptake by mature forest trees. This study investigated the effect of high  $O_3$  events on gross primary productivity (GPP) for a Scots pine stand near Antwerp, Belgium over the period 1998-2013. Stomatal O3 fluxes were modelled using in situ O3 mixing ratio measurements and a multiplicative stomatal model, which was parameterised and validated for this Scots pine stand. Ozoneinduced GPP reduction is most likely to occur during or shortly after days with high stomatal O3 uptake. Therefore, 15 a GPP model parameterised for days with low stomatal O3 uptake rates was used to simulate GPP during periods of high stomatal  $O_3$  uptake. Possible Eventual negative effects of high stomatal  $O_3$  uptake on GPP would then result in an overestimation of GPP by the model during or after high stomatal  $O_3$  uptake events. The  $O_3$  effects on GPP were linked to AOT40 and POD1+. Although the critical levels for both indices were exceeded in every single year, no significant negative effects of O3 on GPP were found and no correlations between GPP residuals and 20 AOT40 and POD1y were found. Overall, we conclude that no O3 effects were detected on the carbon uptake by this Scots pine stand.

**1** Introduction**

25

Tropospheric ozone ( $O_3$ ) is a secondary air pollutant that has the potential to negatively affect vegetation, leading to reduced growth and carbon sequestration potential (ICP Vegetation, 2012;Middleton, 1956). Background concentrations of tropospheric  $O_3$  have increased with 36 % since pre-industrial times (IPCC, 2001) and are projected to further increase considerably until about 2050 (IPCC, 2007). Depending on the scenarios, background  $O_3$  levels might either increase or decrease after 2050 (IPCC, 2007).

In recent years, many studies have been conducted to investigate the mechanisms underlying the O3 impacts on vegetation. Ozone reduces plant growth by altering photosynthetic rates, carbohydrate production, carbon
sequestration, carbon allocation, and carbon translocation (Reich and Amundson, 1985;Andersen et al., 1997;Beedlow et al., 2004). Once O3 enters the leaves through the stomata, it can affect plant growth by direct cellular damage (Mauzerall and Wang, 2001), leading to visible leaf injury and reduced leaf longevity (Noble and Jensen, 1980). In response to O3, respiratory processes increase, which will also affect the tree's carbon balance (Darall, 1989). Skärby et al. (1987) proved that dark respiration of Scots pine shoots increased after long-term

35 exposure to a low level of O3. Protective responses, such as compensation (e. g. repair of injured tissue), avoidance

(e. g. stomatal closure), and tolerance (e. g. alteration of metabolic pathways), all consume carbon and, hence, resistance to  $O_3$  damage costs energy. The size of this cost affects the amount of carbon remaining to support growth (Skärby et al., 1998).

To assess the impact of O3, several indices have been created, e. g. AOT40 (ppb h), the cumulated O3 mixing ratio
in excess of a threshold of 40 ppb, and PODy, the accumulated O3 flux above a flux threshold y (nmol m-2 s-1). Critical levels, quantitative estimates of exposure to O3 above which direct adverse effects may occur (CLRTAP, 2015), have been determined for these indices based on O3 dose-response relationships from fumigation experiments with enhanced O3 mixing ratios (Karlsson et al., 2004). The magnitude of the O3 impact on plants depends on the intensity of O3 exposure, environmental factors influencing both plant photosynthesis and the O3 flux to plant surfaces, and plant species-specific defensive mechanisms (Musselman and Massman, 1999). Because of the variable plant responses to similar O3 mixing ratios, the question arises whether widely applicable tolerable limits of O3 mixing ratio exist (Skärby et al., 1998).

While high stomatal  $O_3$  fluxes have been shown to affect crop yields and the growth of tree seedlings and saplings, it is not sure whether O3 uptake or O3 flux also negatively affects the carbon uptake by mature forest trees. Many 50 studies determined the effect of O3 on seedlings and young trees (e.g. (Buker et al., 2015)), but-little is known about the effect on mature trees. When scaling up the results from seedlings to mature trees the resulting data should be viewed with caution, due to differences in energy budgets, canopy:root balances and architecture and carbon allocation patterns (McLaughlin et al., 2007; Chappelka and Samuelson, 1998). In addition to the uncertainties related with the up-scaling from seedlings to mature trees, data from controlled experiments should also be used 55 with caution, because trees can react differently in field conditions (Skärby et al., 1998). The effect of  $O_3$  uptake on carbon uptake under ambient O3 mixing ratios by trees has hardly been studied in situ. Some studies showed reductions in plant growth due to stomatal O3 uptake (Zapletal et al., 2011;Fares et al., 2013;Yue and Unger, 2013), while other studies did not show any effect (Zona et al., 2014;Samuelson, 1994). Whether or not an effect of stomatal O3 uptake was found was species- and site- specific, and there is a clear need for more studies 60 investigating the effect of O3 on carbon uptake by mature trees in the field (Chappelka and Samuelson, 1998).

In this studyHere we investigated the effect of high O3 events at ambient levels on gross primary productiovityn (GPP) for of a 
[revised manuscript text omitted]

(1)

175 where
$$R_{u1}$$
 is the total resistance to  $Q_{u}$ .  $R_{un}$  is the aerodynamic resistance to  $Q_{u}$ .  $R_{u1}$  is the quast-laminar boundary layer resistance to  $Q_{u}$  and  $R_{un}$  is the canopy resistance to  $Q_{u}$  (all expressed in s m2).
 $g_{mnr} = \left(\frac{\lambda}{m_{unr}} + \frac{\lambda}{m_{u1}} + \frac{\lambda}{m_{u1}}\right)^{\frac{1}{m_{u1}}}$  (2)
where  $g_{un}$  is the total conductance to  $Q_{u}$  ( $g_{uur}$  is the canopy conductance.
180 The aerodynamic conductance to  $Q_{u}$  ( $g_{uur}$  is the canopy conductance.
180 The aerodynamic conductance to  $Q_{u}$  ( $g_{uur}$  is the canopy conductance.
180 The aerodynamic conductance to  $Q_{u}$  ( $g_{uur}$  is the fiction velocity. L is the obview length  $z$  is the height
 $g_{unrev} = \frac{1}{m_{u1}} \left[ \ln \left(\frac{z-z}{m_{u1}}\right) + \Psi_{n} \left(\frac{z-z}{m_{u1}}\right) + \Psi_{n} \left(\frac{z-z}{m_{u1}}\right) \right]$  (1)
where the von Karman constant  $z = 0.41$ ,  $u^{2}$  ( $m \in 1$ ) is the fiction velocity. L is the obview length  $z$  is the height
 $at$  which the  $Q_{u}$  mixing ratio was measured;  $d$  is the zero plane displacement (Appendix C),  $z_{u}$  is the momentum
reagineer, parameter (Appendix C). Whit the atmospheric stability function calculated using the set of coefficients
published by Dyer (1974) and is described in detail in Appendix C.
The aerodynamic resistance was calculated following (Grinhage, 2002) with:
 $R_{urrov} = \frac{1}{m_{u1}} \left[ \ln \left(\frac{z-d}{m_{u}}\right) - \frac{1}{m_{u}} \left(\frac{z-d}{m_{u}}\right) + \frac{1}{m_{u}} \left(\frac{z-d}{m_{u}}\right) \right]$
 $\frac{1}{m_{u1}} \left[ \ln \left(\frac{z-d}{m_{u}}\right) - \frac{1}{m_{u}} \left(\frac{z-d}{m_{u}}\right) + \frac{1}{m_{u}} \left(\frac{z-d}{m_{u}}\right) \right]$
 $\frac{1}{m_{u1}} \left[ \ln \left(\frac{z-d}{m_{u}}\right) - \frac{1}{m_{u}} \left(\frac{z-d}{m_{u}}\right) + \frac{1}{m_{u}} \left(\frac{z-d}{m_{u}}\right) \right]$
 $\frac{1}{m_{u2}} = \frac{1}{m_{u1}} \left[ \ln \left(\frac{z-d}{m_{u}}\right) + \frac{1}{m_{u}} \left(\frac{z-d}{m_{u}}\right) \right]$
 $\frac{1}{m_{u2}} = \frac{1}{m_{u1}} \left[ \ln \left(\frac{z-d}{m_{u}}\right) + \frac{1}{m_{u}} \left(\frac{z-d}{m_{u}}\right) \right]$
 $\frac{1}{m_{u2}} = \frac{1}{m_{u2}} \left[ \ln \left(\frac{z-d}{m_{u}}\right) + \frac{1}{m_{u}} \left(\frac{z-d}{m_{u}}\right) \right]$
 $\frac{1}{m_{u2}} = \frac{1}{m_{u2}} \left[ \ln \left(\frac{z-d}{m_{u}}\right) + \frac{1}{m_{u1}} \left(\frac{z-d}{m_{u2}}\right) \right]$
 $\frac{1}{m_{u2}} = \frac{1}{m_{u2}$

where  $\kappa$  is the von Karman constant (0.43); u\* (m s-1) is the friction velocity, which is derived from the measured momentum fluxes; Sc is the Schmidt number (1.07 for O3); Pr is the Prandtl number (0.72 for O3); 44.64 mol m-3 is the molar density of air (at an air pressure of 101.3 kPa and an air temperature of 0°C), and is applied for converting the unit of gbl from m s-1 to mol m-2 s-1.

The quasi-laminar boundary layer resistance was calculated following (Baldocchi et al., 1987) with:

205

210

230

$$R_{bl} = \frac{2}{\kappa * u^*} \left(\frac{Sc}{Pr}\right)^{2/3}$$
(9)

where  $\kappa$  is the von Karman constant (0.43), u\* (m s-1) is the friction velocity, which is derived from the measured momentum fluxes, Sc is the Schmidt number (1.07 for O3), and Pr is the Prandtl number (0.72 for O3).

The canopy conductance consisted of a stomatal and a non stomatal component. Since the stomatal component varies throughout the canopy, the canopy was divided into eight sublayers so that the leaves were evenly distributed between the horizontal layers. Dividing the canopy into sufficient sublayers was necessary in order to model fluxes well. Eight sublayers were considered to be sufficient, as indicated in a sensitivity test with more and less sublayers.

215 For each leaf layer, the model calculates  $g_{st}$  for sunlit and shaded needles, taking the solar elevation angle into account. Non stomatal conductance was assumed to be constant over the canopy and was set to 0.16 mol m-2-s-1. This value was derived from long term O3 flux measurements in Brasschaat (Neirynck et al., 2012).

The stomatal and non stomatal  $O_{\underline{3}}$  fluxes (nmol m-2 s-1) were calculated by multiplying the proportion of  $g_{\underline{st}}$  and  $g_{\underline{ns}}$  of the canopy per ground area with the  $O_{\underline{3}}$  mixing ratio.

[revised manuscript text omitted]

The model calculates half hourly totals of the total, stomatal, and non-stomatal O3-fluxes based on the following input variables: day of year, hour, Rg, Tuit, VPD, SWP, O3-mixing ratio above the canopy (24m), LAI, and friction velocity u\*. The total O3-flux (nmol m-2-s-4) for the whole canopy was the product of O3-mixing ratio (ppb) and gtot (mol (m2 ground area-4) s-4) (Musselman and Massman, 1999). This last parameter was calculated with an electrical model (Eq. 2).

$$g_{tot} = \left(\frac{1}{g_{tot}} + \frac{1}{g_{tot}} + \frac{1}{g_{tot}}\right)^{-1} \tag{2}$$

where  $g_{tot}$  is the total conductance to  $\Theta_3$ -(mol (m2 ground area-4) s-4);  $g_{acro}$  is the aerodynamic conductance;  $g_{bl}$  is the boundary layer conductance to  $\Theta_3$ ;  $g_{can}$  is the canopy conductance.

**285**

290

275

280

The aerodynamic conductance gaero was calculated with the following formula (Grünhage, 2002):

$$g_{\frac{dero}{dero}} = \frac{\pm}{\kappa u^{\pm}} \left[ \ln\left(\frac{z-d}{z_{\overline{a}}}\right) - \Psi_{\overline{a}}\left(\frac{z-d}{\pm}\right) + \Psi_{\overline{a}}\left(\frac{z_{\overline{a}}}{\pm}\right) \right] \tag{3}$$

where the von Karman constant  $\kappa = 0.43$ ; u\* (m s 1) is the friction velocity; L is the obukov length; z is the height at which the O3-mixing ratio was measured ; d is the zero plane displacement (Appendix C); z0 is the momentum roughness parameter (Appendix C); Wh is the atmospheric stability function calculated using the set of coefficients published by Dyer (1974) and is described in detail in Appendix C.

The boundary layer conductance to O2 was calculated with the following formula (Baldocchi et al., 1987):

$$g_{\theta\theta} = \frac{\frac{2}{2} + \frac{2}{2} + \frac{2}{p_{T}}}{\frac{2}{p_{T}} + \frac{2}{p_{T}}} + \frac{44.64}{p_{T}}$$
(4)

where  $\kappa$  is the von Karman constant (0.43); u\* (m s-4) is the friction velocity, which is derived from the measured momentum fluxes; Sc is the Schmidt number (1.07 for O3); Pr is the Prandtl number (0.72 for O3); 44.64 mol m-3 295 is the molar density of air (at an air pressure of 101.3 kPa and an air temperature of 0°C), and is applied for converting the unit of gbl from m s4-to-mol m2 s4.

The canopy conductance consisted of a stomatal and a non-stomatal component. Since the stomatal component varies throughout the canopy, the canopy was divided into eight sublayers so that the leaves were evenly distributed between the horizontal layers. Dividing the canopy into sufficient sublayers was necessary in order to model fluxes well. Eight sublayers were considered to be sufficient, as indicated in a sensitivity test with more and less sublayers.

For each leaf layer, the model calculates g# for sunlit and shaded needles, taking the solar elevation angle into account. Non-stomatal conductance was assumed to be constant over the canopy and was set to 0.16 mol m-2 s-1. This value was derived from long-term O2 flux measurements in Brasschaat (Neirynek et al., 2012).

[revised manuscript text omitted]

whereHere gst is the stomatal conductance to O3 and gmax is the species specific maximal stomatal conductance to O3. The other parameters are expressed in relative terms as a proportion of gmax; fmin is the ratio of gmin to gmax; where gmin is the minimal stomatal conductance that occurs during daylight period; fphen represents the modification of gmax, due to phenological changes; fPAR represents the modification of gmax by photosynthetically active radiation (PAR); fT represents the modification of gmax by vapour pressure deficit (VPD); fSWP represents the modification of gmax by soil water
potential (SWP). The functions fPHEN, fPAR, fT, fVPD, and fSWP represent the modification of gmax where gmin is the minimal stomatal conductance to O3. 
[revised manuscript text omitted]
_{bF}$  is the boundary layer conductance;  $g_{ean}$  is the canopy conductance.

 $g_{aero} = \frac{1}{\kappa u^{\pm}} \left[ \ln(\frac{z-d}{z_{0}}) - \Psi_{h}\left(\frac{z-d}{L}\right) + \Psi_{h}\left(\frac{z_{0}}{L}\right) \right]$ (C2)

660 where the von Karman constant  $\kappa = 0.43$ ; u\* (m s 1) is the friction velocity; L is the obukov length; z is the height at which the O3 mixing ratio was measured ; where  $z_{nm} = 0.1 * h$  and d = 0.65 \* h with h is the canopy height; and where the atmospheric stratification function  $\Psi_{h}$  is calculated as: • Unstable atmospheric stratification (L < 0m):  $\Psi_{h} = 2 * \ln[\frac{1}{\varphi_{h}(\zeta)} + 1]$ 665 with  $\varphi_{\mu} = (1 - 16 * \zeta)^{-0.5}$ and  $\zeta = \frac{z-d}{L}$  with  $z = z_2 = z_{ref,T}$  and  $z = z_1 = d + z_{om}$  and L = the Obukov length • Stable atmospheric stratification (L > 0m):  $\Psi_{\rm p} = -5 * \zeta$ with  $\zeta = \frac{z-d}{L}$  with  $z = z_2 = z_{ref,T}$  and  $z = z_1 = d + z_{0m}$  and L = the Obukov length 670 • Neutral atmospheric stratification ( $|L| \rightarrow \infty$ ):  $\Psi_{n} = 0$ Total O3 flux (nmol m-2 ground area s-1) is the O3 flux for the whole canopy and is then calculated by:  $F_{tot} = [O_3] * g_{tot}$ (C3) 675 where  $[O_3]$  is the  $O_3$  mixing ratio (ppb). Stomatal  $O_3$  flux is the fraction of the total  $O_3$  flux taken up by the stomata and is calculated by:  $F_{st} = F_{st} * \frac{g_{st}}{g_{st} + g_{ms}}$ (C4) where gst is the stomatal conductance (mol m-2 ground area s-1); gns is the non-stomatal conductance (mol m-2 ground area s-1). 680 Non-stomatal  $O_3$ -flux ( $F_{ts}$ ) is the difference between total  $O_3$ -flux and stomatal  $O_3$ -flux:  $F_{ns} = F_{tot} - F_{st}$ (C5)Part 2 The solar elevation submodel This submodel calculates the solar elevation angle,  $\beta$  (radians), at each time step (Campbell and Norman, 1998).  $\beta = \arcsin\left(\sin\phi\sin\delta + \cos\phi\cos\delta\cos h\right)$ 685 -(C6)22

|     | where $\delta$ is the solar declination angle; $\delta = -23.4 * \left(\frac{\pi}{180}\right) * \cos(2 * \pi * \frac{doy+10}{365})$                                   |
|-----|-----------------------------------------------------------------------------------------------------------------------------------------------------------------------|
|     | $\phi$ is the latitude in radians; $\phi = 0.89$                                                                                                                      |
|     | h is the hour angle of the sun; $h = \pi * \frac{t - t_{g}}{12.0}$                                                                                             |
|     | where t is time; $t_0$ is solar noon; $t_0 = 12 + \frac{4*(L_s - L_\theta) - E_t}{60.0}$                                                                              |
| 690 | $L_s$ is the standard longitude in degrees; $L_s = 15.0$                                                                                                              |
|     | $L_t$ is the local longitude in degrees; $L_t = 4.0$                                                                                                                  |
|     | $E_{\varepsilon}$ is the empheris of the sun;                                                                                                                         |
|     | $E_{\overline{e}} = 0.017 + 0.4281 * \cos(F_{\overline{e}}) - 7.351 * \sin(F_{\overline{e}}) - 3.349 * \cos(2 * F_{\overline{e}}) - 9.731$ $* \sin(F_{\overline{e}})$ |

[revised manuscript text omitted]

875 Table 2. Performance statistics for the multiplicative stomatal model: mean bias (MB), relative mean error (RME), systematic and unsystematic root mean squared error (RMSEs/u), Willmott's index of agreement (d), model efficiency (ME), coefficient of determination (R2).

| Statistics        | Parameterisation | Validation |
|-------------------|------------------|------------|
| MB                | 0.002            | 0.002      |
| RME               | 0.34             | 0.33       |
| RMSE              | 0.019            | 0.019      |
| RMSE s | 0.006            | 0.006      |
| RMSE u | 0.017            | 0.017      |
| d                 | 0.99             | 0.99       |
| ME                | 0.72             | 0.72       |
| R 2    | 0.72             | 0.72       |

880 Fig. 1. Fingerprint of air temperature (Tair), incoming global radiation (Rg), vapour pressure deficit (VPD), and measured gross primary productivity (GPP), averaged over the period 1998-2013. Day of year is plotted on the y-axis and hour of day on the x-axis.

---

## Referee Report (RR1)

This is my second review of this paper. In the re-submitted version the authors substantially improve the readability of the text. They have considered my initial comments, deepening the methods section and discussing the possible confounding effect of solar radiation on the results. The latter was investigated comparing the frequency distribution of meteorological variables such as solar radiation, temperature and VPD (figure 6) for the dataset on which they assumed an O3 effect over GPP and the dataset on which an O3 effect over GPP is not expected.

Histograms reported in figure 6 show that the frequency distributions of meteorological variables in the two dataset are not the same. The discrepancy between the two distributions is strongly noticeable for solar radiation (figure 6 a), confirming my initial concern on the possible confounding effect of this variable. Figure 6 a shows that solar radiation is usually high during high O3 events, so that a model parameterized under low O3 condition would be also parameterized under low irradiation, leading to a GPP underestimation that would mask a possible negative effect of O3 over GPP.

In my opinion, the main conclusion, stated as "No O3 effects were detected on the carbon uptake by the Scots pine stand", is not supported by a valid method, which need to be revisit.

However this work is interesting , a good amount of data are presented and I believe that the paper could be a useful addition to the literature of O3 forest damage if authors take on such an effort in revisiting the data analysis in order to avoid confounding effect of meteorological variables. I strongly encourage authors to resubmit it.

---

## Editor Decision (ED1)

**Some specific comments to:**

**bg-2016-12**

**No tropospheric ozone impact on the carbon uptake by a Belgian pine forest**

By I. Trebs

In general, $O_3$ with the unit ppb should be referred to as mixing ratio instead of concentration. This should be changed throughout the manuscript.

It is unclear to the referee, why the aerodynamic conductance could not be calculated from the data retrieved by the eddy covariance measurements (e.g., $u_*$)? I think a multi-layer approach that considers all conductances of the canopy layers (also the in-canopy aerodynamic conductance) would be more appropriate than a big-leaf approach, but therefore a measured or known vertical profile of the $O_3$ mixing ratio (and wind speed) has to be known.

Equation 3: Please introduce a symbol for the molar density of air.

Line 195 should read: where $\kappa$ **is the von Kármán** constant (replace K with the Greek letter also in the formula)

Line 197: molar density of air → at which temperature (and pressure)?

Line 203: Non-stomatal conductance was assumed to be constant over the canopy and was set to 0.16**…units are missing here (mol m$^{-2}$ s$^{-1}$)**.

Text and Figures: please double check especially the figures, units of $O_3$ flux density must be **nmol m$^{-2}$ s$^{-1}$** and not mol m$^{-2}$ s$^{-1}$.

Information of the canopy height is missing.

**APPENDIX C, equation C3:**

This equation appears to be wrong. The mixing ratio of $O_3$ is multiplied by a dimensionless factor given by a ratio of conductances. The resulting unit would be a mixing ratio and not a flux density with the unit nmol m$^{-2}$ s$^{-1}$.

The conventional equation to retrieve $F_{st}$ is (e.g. Gerosa et al., 2005):

$$F_{st} = g_{st} \cdot O_3(canopy)$$

Here, $O_3$ (canopy) is the $O_3$ mixing ratio just above the canopy top or at the zero plane displacement height. Note, that this mixing ratio should not be identical to the one used in equation C2 (which is the $O_3$ mixing ratio at the measurement height).

**Obviously, the overall results of the manuscript have to be revised if equation C3 was indeed used to derive $F_{st}$. The conclusions may change accordingly.**

---

## Author Response (AR2)

Dear Editor,

We are pleased to resubmit the revised version of our manuscript "No impact of tropospheric ozone on the gross primary productivity of a Belgian pine forest". We have addressed each of the concerns of both referees as outlined below.

**Referee #2**

**The referee's comment:** 1. I think that the ultimate sentence about ozone implication in altering the NPP/GPP ratio (lines 410-415) needs to further work to be well ascertained.

**The authors' answer:**

We agree with the referee that further research is needed to investigate the effect of $O_3$ on growth, NPP, NPP/GPP ratios and other processes in the Scots pine stand. We have slightly reformulated the last paragraph.

**The referee's comment:** 2. Some minor comments:
- Paragraph 2.2.4 Gross Primary Production (lines 131-132). Authors wrote "Gross primary productivity was derived by subtracting the modelled total (autotrophic and heterotrophic) ecosystem respiration from the measured NEE". I think that:

NEE = NPP-Rh = GPP-(Ra + Rh) = GPP-Reco

thus GPP = NEE + Reco

where NEE: net ecosystem exchange (measured), NPP: Net Primary Production, Ra: Autotrophic respiration; Rh: Heterotrophic respiration, Reco: ecosystem respiration (total respiration).

**The authors' response:**

In this study GPP was always positive, and we made thus use of the convention stating that a downward flux is a positive flux. Therefore, we agree with the referee that GPP = NEE + Reco. We adapted this in the manuscript (Appendix A).

- Figure 7. Please, check regression equation of data concerning the panel (A): intercept is set at 7779 whereas in panel (B) is set at -0.0658 with similar angular coefficients (slopes) and determination coefficients.

**The authors' response:**

The regression equations on both panels are correct. We adapted the scale of this figure in accordance with figure 8 and added this new figure to the manuscript.

[Figure]

- Please check all corresponded references between text and bibliographic list.

**The authors' response:**

We thank the referee for this remark. Due to a problem with endnote, some references did not show up in the reference list or were misspelled. We solved this problem.

**Referee #5**

**The referee's comment:** 1. GPP is the key parameter of this paper and should be introduced in more detail. Why did the authors choose GPP as the effect parameter, how is it linked to tree/forest growth, timber production, C sequestration etc.? Why is the effect of ozone on GPP of (ecological/economic) interest? Is there evidence in the literature that ozone affects (tree) GPP?

**The authors' response:**

The study of $O_3$ effects on GPP is relevant because GPP represents the first step in the process of C assimilation and with GPP we quantify the rate at which C substrate is provided for growth, wood production, et cetera. There is evidence in literature that $O_3$ affects GPP. Fares et al. (2013) investigated the effect of $O_3$ on C assimilation in trees. They found that $O_3$ has a direct negative effect on C assimilation by plants, and that this negative effect of $O_3$ on GPP mainly occurred within a day of exposure/uptake.

**The referee's comment:** 2. Due to the importance of GPP in this study, the measurement of NEE in this forest stand and the derivation of GPP from measured NEE have to be described in much more detail, at least in the appendix (the sentence "…(NEE) measured with the eddy covariance technique with the instrumentation and following the standard data quality procedures as explained in (Carrara et al., 2003;Carrara et al., 2004;Gielen et al., 2013)." is not sufficient and too vague (e.g., which instrumentation?)). Also, for which time period was NEE measured/GPP derived? For the entire

growing season of all 14 years? If so, why don't the authors show in the results section scatterplots of GPP (as derived from measured NEE) against Fst or POD1?

**The authors' response:**

We have added a more detailed description of the NEE measurements and the GPP derivation in an Appendix (Appendix A).

Measurements of NEE started in 1998 (and thus GPP derived). Data from 1999 and 2003 were removed from the dataset due to poor data quality or coverage. It is now clearly mentioned in paragraph 2.2.4 for what period GPP values were derived and integrated.

The figure below (Figure 1) on the left shows the scatter plots of measured growing season GPP versus $F_{st}$, AOT40 and POD1. We have considered including these scatter plots in an earlier version of the manuscript, but found it more relevant to show GPP residuals. We think that plotting GPP versus $O_3$ doses is not very meaningful, because GPP variation between growing seasons is determined more by climatic variability or LAI variability than by $O_3$ loads. If the referee deems it necessary, we are nevertheless willing to include the scatter plots in the manuscript, e.g. as part of Figure 9.

*Remark:* The scatter plots show a negative relationship between GPP and $O_3$ dose, most notably between GPP and $F_{st}$ (upper panel). These trends suggest a strong effect of $O_3$ on GPP - in the upper panel in the lower range of $F_{st}$ (but then strangely no further effect at higher $F_{st}$ values). The figure below on the right shows the time series of GPP and AOT40. On these time series it can be seen that there is a period of steadily increasing GPP, from 2006 until 2013, which happens to coincide with a decreasing trend of $O_3$ concentrations (AOT40). The increasing GPP involves forest recovery from acidification (Neirynck et al., 2008). The negative relation between GPP and $O_3$ dose for this period might wrongly be interpreted as an $O_3$ effect.

[Figure]

*Figure 1. (left) The measured growing season GPP in function of Fst, AOT40 and POD1, (right) GPP and AOT40 in function of the years of the measuring period of this study.*

**The referee's comment:** 3. For the derivation of LAImax across all years, measurements were also taken from the year 2003 (weather-wise an extreme year that unfortunately had to be excluded from this study – it would have been very interesting to see if it had created an outlier in figures 8 (E,F) and 9): How sure are the authors that this 2003 measurement was representative, given that European forests suffered immensely during that year from heat and drought, which might have had an effect on LAI?

**The authors' response:**

The LAImax of 2013 was 1.67 m² m⁻², which is higher than the other years after the thinning. We assume this value was representative because the dry period in 2003 reached its peak in August, after the seasonal LAI maximum had been reached and was measured (half July). The interannual variability after the thinning in 1999 was rather small, thus LAI is no big driver in stomatal uptake differences between years.

**The referee's comment:** 4. It would have been interesting to do a year-to-year comparison of ozone effects on GPP to analyse whether the timing/onset/length of ozone episodes (e.g. late/mid/end of

season) had any effect on GPP; the authors could for example add the year to data points in Fig. 9 and relate outlier years in these figures to ozone concentration anomalies in these years?

**The authors' response:**

In the first version of this manuscript we included a figure showing the GPP residuals of the growing season against Fst, POD1 and AOT40 for the different years (Figure 2). We replaced this figure by Figure 9 in our manuscript to improve the readability, as the initial figure with different colours for each year did not gave any extra information and could confuse readers.

[Figure]

*Figure 2. GPP residuals in relation to the stomatal $O_3$ uptake (Fst). The different colours represent the different years over the study periods.*

Here we show the total stomatal $O_3$ uptake for each week for the positive anomaly, neutral and negative anomaly in Figure 9 of the manuscript (Figure 3). The highest positive anomaly in GPP residuals was reached in 2011 and 2012, whereas the most negative anomaly was in 2010 and 2013. In 1998, 2004, 2007, 2008 and 2009 the Fst values are comparable to 2010-2013, but no anomaly was found. In 2002 the GPP residuals are close to zero, but a high Fst was reached. In the weekly Fst data we don't see any clear differences between the four panels.

[Figure]

*Figure 3. The stomatal O₃ uptake (Fst) for each year, for 2002 (no anomaly in Fig. 9 of the manuscript, but reaching high Fst values), 2011-2012 (positive anomaly in Fig. 9), 1998-2004-2007-2008-2009 (no anomaly in Fig. 9), 2010-2013 (negative anomaly in Fig. 9).*

**The referee's comment:** 5. The description of the used ANN is very short, please add detail how the final model was derived. Is there any particular reason why the authors used ANNs for the analysis (could the authors maybe justify with reference to other biological studies that also used ANNs?)? One main strength of ANNs is their power to disentangle strong and weak input-output relationships, so one would have expected to at least see a ranking of parameters that define GPP. For ANN model runs with the full dataset (i.e. no exclusion of high ozone GPP days), the authors should have also included ozone as input to see whether it has any explanatory power on GPP.

**The authors' response:**

ANN's are a power tool to process multidimensional data in which complex nonlinear interrelationships between the parameters can be expected. In this manuscript we don't know the exact relationships between the meteorological parameters and GPP. It would lead us too far to determine these exact relations and the ANN offers us a great tool to overcome this problem. ANN's are used in terrestrial and aquatic ecosystems, remote sensing and evolutionary ecology (Lek and Guegan, 1999; Akhand et al., 2016; Liu et al., 2016), and specifically to model GPP (Rochelle-Newall et al., 2007).

We ranked all the input parameters of our model by replacing each input variable with a random permutation of its values (Table 1; Table 3 in the manuscript). Global radiation (Rg) contributed most to GPP, followed by year and day of year. Soil water content (SWC) contributed the least.

When we included O₃ as input variable to this model and calculated its contribution to GPP by ranking the parameters, O₃ has the lowest MSE. We concluded that O₃ does not contribute to the prediction of GPP in our model.

This information was added in the manuscript at the appropriate text locations in Introduction, in M&M paragraph 2.5 Detecting O₃ effects on GPP), the Results in paragraph 3.3 Ozone effects on GPP, and in Discussion in paragraph 4.3 Ozone effects on GPP.

*Table 1. Ranking of the parameters that define GPP in the ANN by replacing each input variable with a random permutation of its values. (A) The parameters with their mean squared error (MSE, mol m$^{-2}$ day$^{-1}$) for the model without O$_3$ (B) The parameters with their MSE for the model with O$_3$. The overall model MSE without any random permutation is also shown.*

| Ranking Nr. | A | B |
|---|---|---|
| 1 | Rg – 37500.81 | Rg – 41358.93 |
| 2 | doy – 30240.61 | year – 33978.09 |
| 3 | year – 27486.63 | doy – 31127.90 |
| 4 | VPD – 15380.68 | Tsoil – 24893.78 |
| 5 | Tmax – 15323.22 | Tmax – 23567.45 |
| 6 | Tsoil – 15076.75 | Tmean - 21354.76 |
| 7 | Tmean – 13858.91 | VPD – 16395.14 |
| 8 | WV – 13369.01 | Tmin – 15418.16 |
| 9 | Tmin – 12732.96 | WV – 14685.97 |
| 10 | SWC – 12402.04 | SWC – 12831.19 |
| 11 | | O$_3$ – 11885.73 |
| | | |
| Overall model MSE | 11360.85 | 10019.30 |
| | | |

**The referee's comment:** 6. Line 204f.: "Under the assumption that O3-induced 205 GPP reduction is most likely to occur during and shortly after days of high stomatal O3 fluxes, ..", Line 239ff. "now not only excluding the days with the highest stomatal O3 fluxes from the dataset for model training but also the two following days." And Line 255f. "we excluded the days with a peak of stomatal O3 flux plus the six following days". The reasoning behind these timeframes is rather vague and these are big assumptions that need scientific discussion: Is there experimental/statistical evidence that high ozone episodes only have an effect on GPP for up to two days after the episode? Why exactly six days to look at carry-over effects? Surely timing will depend on the intensity and length of the ozone episode? How do the authors distinguish between "high" and low ozone episode? Where did they draw the line (please show how many high and low ozone episodes there were in each year)? The authors call the latter episodes "supposedly O3-damage free", which is a very bold statement. This is one of the weakest point of the methodology that should be thoroughly addressed in the revision!

**The authors' response:**

Firstly, we have added a few lines in paragraph 2.5 to explain the physiological mechanism on which our assumption is based: above a certain O$_3$ load, the tree's defensive mechanisms cannot scavenge all O$_3$ entering the needles and damage to the photosynthetic apparatus occurs. This leads to decreased gross photosynthetic rates and GPP. The damage is repaired afterwards when the O$_3$ load decreases.

Because we lack information on the defensive capacity of the trees and on the rate at which trees are able to repair O$_3$ damage to the photosynthetic apparatus, we performed and repeated our analysis (1) with three different peak percentages of fluxes to define the days at which an O$_3$ effect might occur and (2) with three different delay-periods after a peak flux that an O$_3$ effect can sustain. The latter was not mentioned in the manuscript, but we have done the tests with a 1-day, 2-day and 6-day delay period. Because the outcome of the analysis was similar amongst the three delay periods, only the results for the 2-day period are shown in the manuscript. We are willing to show the other results to the referee if he acts this to be necessary.

The three peak percentages and the three delay-periods have both been selected rather arbitrarily as a sample from a whole range of thresholds and delay-period lengths that could have been tested in the rather empirical approach we apply in this study to identify days with and without $O_3$-affected GPP.

To respond to the last part of the referee's comment: we made a distinction between high and low $O_3$ episode applying a stomatal $O_3$ flux treshold of 0.18 mmol $O_3$ m$^{-2}$ day$^{-1}$. This is a 10 % cut-off. To make an $O_3$-damage free GPP model this cut-off value was the maximum value we tested for the days on which we expected a possible $O_3$ effect. In the table below we show per year the amounts of days which were above this value.

*Table 2. The amount of days which have an Fst value > 0.18 mmol m$^{-2}$ day$^{-1}$ shown per year. These were considered as 'high $O_3$ episode'.*

| Year | Amount of days with high Fst |
|------|------------------------------|
| 1998 | 41 |
| 2000 | 13 |
| 2001 | 31 |
| 2002 | 51 |
| 2004 | 15 |
| 2005 | 18 |
| 2006 | 33 |
| 2007 | 17 |
| 2008 | 23 |
| 2009 | 16 |
| 2010 | 8 |
| 2011 | 1 |
| 2012 | 7 |
| 2013 | 13 |

**The referee's comment:** 7. It is surprising that the authors chose the Jarvis-type stomatal conductance model rather than the Ball-Berry photosynthesis-based stomatal conductance model, which would have offered a more direct link between ozone uptake and photosynthesis/GPP. Please add a few sentences on this type of model to discussion section.

**The authors' response:**

We have considered using a Ball-Berry-type stomatal model coupled with Farquhar's biochemical photosynthesis model. As the referee correctly states, we could with this model simulate photosynthesis rates/GPP in addition to the stomatal $O_3$ fluxes. However, if dose-response relations are investigated as we do in our study (the response = GPP or GPP residual), it is crucial that the response is measured or quantified entirely independently from the dose. This is not the case if we involve the photosynthesis rates/GPP simulated with the coupled photosynthesis-Ball-Berry-type model in the calculation of the response. In this regard, the photosynthetic rates/GPP that we could obtain with that model are of little use in our study.

We then preferred the Jarvis-type stomatal model over the Ball-Berry-type model because we want to publish and provide to the ozone risk assessment community a full site-specific parameterisation of the model for Scots pine. As the referee probably knows, the Jarvis type model is embedded in the DO$_3$SE algorithm that is currently used to calculate ozone doses for European forests (Mapping

Manual).Scots pine is the representative species for coniferous forests in Atlantic Central Europe and the parameterisation of the stomatal model for Scots pine is based on a compilation of primary and secondary data. We like to publish a full parameterisation of the model for a specific Scots pine stand in the region, with which the DO$_3$SE parameterisation can be compared and which may possibly contribute to an improved parameterisation. In paragraph 4.1 we added some lines in which we compare the two parameterisations.

**The referee's comment:** 8. The authors should discuss long-term effects of ozone on forest/tree productivity (e.g. over the entire length of the 14 year study period), which presumably can't be captured by the approach chosen?

**The authors' response:**

In this study we focus on short term effects of O$_3$ on GPP. Tree ring growth data and wood density data are available for this forest. It will be interesting to test if O$_3$ has a long term effect on the biomass, but this is beyond the scope of this research. These data will be analysed in a follow-up paper.

**The referee's comment:** 9. Related to 8: Please more clearly summarise the limitations of your study/approach.

**The authors' response:**

We have added some lines to the last section of the Discussion where we summarize the limitations of the method we apply, in particular the distinction between days with and without an O$_3$ effect on GPP.

**The referee's comment:** 10. In Appendix A the authors introduce the stomatal model, including fphen, which is then set to 1 in the model runs, right? So maybe remove detail in appendix? Also, how was the start and end of growing season identified? Given the change in weather, one would have expected a variation of these two parameters from year-to-year.

**The authors' response:**

The function $f_{phen}$ was not set to 1 in the model runs. The function was set to 1 only for the parameterisation of the model against the in situ $g_{st}$ measurements, because these measurements had all been collected on fully matured needles. We rephrased this in paragraph 2.4 so that this is clear to the reader.

The start and end of the growing season were for all years fixed to doy 115 and doy 300, respectively. These values were derived from the LAI course measured in 2007 and taken as the start of LAI increase in spring and the end of the LAI decrease in autumn. There is no doubt that the start and end of the season vary from year to year and this, probably, within a two-week range. The effect of the uncertainty in the start and end of the growing season has only a small effect on the modelled stomatal conductance and stomatal O$_3$ fluxes, though. This is because the phenology function is set to start from and end at a value of 0.8, not a value of 0 or 0.2 or so. Being a canopy parameter, this

value of 0.8 makes sense: at the start of the growing season the pine canopy bears mostly fully grown one-year-old needles and at the end of the growing it bears a mixture of fully-grown current-year needles and partly senescing one-year-old needles. A value of 0.8 is thus meaningful for our pine forest (this in contrast with a deciduous forest, for which $f_{phen}$ should start and end at a very low value, if not 0).

**The referee's comment:** 11-14.

Line 340f.: Please add reference for other Scots pine forests

Some of the references seem to be rather outdated, e.g. line 25 (IPCC, 2001). Please check manuscript for more recent ones.

Please check reference list: some references mentioned in main text are missing (e.g. CLRTAP, 2015; Mills et al., 2011), some authors misspelled (e.g. Büker rather than Buker)

Please check paragraphs starting on line 48 and 204 for spelling. Also, check entire manuscript for BG criteria for correct referencing, e.g. line 124 "explained in (Carrara et al., 2003;Carrara et al., 2004;Gielen et al., 2013)" should be "explained in Carrara et al. (2003, 2004) and Gielen et al. (2013)". Make sure to leave space after ";" when listing references.

**The authors' response:**

We thank the referee for remarking the mistakes with the references. Due to a problem with endnote, some references did not show up in the reference list or were misspelled. We solved this problem.

At line '340' we added two references for other Scots pine forests (Altimir et al., 2004; Emberson et al., 2007). In that paragraph we have also expanded a bit on the differences between our parameterization of the Jarvis model and the parameterization used in the DO$_3$SE model to simulate stomatal O$_3$ doses for Scots pine in the Atlantic Central Europe, as explained in the mapping manual of CLRTAP.

The following references were replaced by more recent ones:

- (IPCC, 2001)
- (Middleton, 1956)
- (Darall, 1989)
- (Noble and Jensen, 1980)
- (Chappelka and Samuelson, 1998)
- (Andersen et al., 1997)

The following references were added to this manuscript:

- (Subramanian et al., 2015)
- (Young et al., 2013)
- (Ashmore, 2005)
- (Wittig et al., 2009)
- (Li et al., 2016)
- (Ainsworth et al., 2012)
- (Huttunen and Manninen, 2013)

[revised manuscript text omitted]

---

## Author Response (AR3)

Dear Editor,

We are pleased to resubmit the revised version of our manuscript after minor revisions. Here are our point-by-point responses to each of the comments of referee #5.

**Referee #5**

**The referee's comment: 1.1** Former comment #2: I am pleased to see that the methods section has been detailed further with help of an appendix. I do think that showing (maybe in the supplement) a revised Figure 1 (as given in your recent response to reviewers) would be of interest to the reader, i.e. how measured GPP relates to measured AOT40 and - much more importantly - Fst and POD1 over time. But rather than showing growing season GPP means (they can indeed lead to a misinterpretation), it would be interesting to see for example daily values of GPP vs. POD1 up until that moment or the ozone dose of the preceding days; in other words, is there a visible effect of (acc.) ozone flux on measured daily GPP? I accept that this might be too cumbersome for this paper, but would like to mention it at least to show that your excellent set of measured NEE/GPP data might hold a lot of very useful additional information on the accumulated and/or instantaneous effect of ozone on forest growth, which potentially contradicts the main statement of this paper.

**The authors' response:**

We show here below the requested plots of measured daily GPP as function of the seasonal accumulated $F_{st}$ (Figure 1) and as function of the $F_{st}$ accumulated over the 5 preceding days (Figure 2). As can be seen on these graphs, GPP increases with accumulated $F_{st}$ in the low Fst range - these are data from days in the early and late growing season - and then rather stagnates in the middle and high Fst range. There is no GPP decrease with increasing $F_{st}$ detectable on these graphs and from this we can infer that there is no visible effect of accumulated ozone dose on measured daily GPP.

We prefer not to include these two figures in the paper, because we share the opinion of the referee that including them would make the paper too "cumbersome". We would however like to include in the supplement the figure that shows measured growing season GPP vs $O_3$ loads (see Figure 3 further below).

[Figure]

*Figure 1. Measured daily GPP (μmol m⁻² day⁻¹) as function of the stomatal O₃ flux (nmol m⁻²) accumulated over all preceding days of the growing season.*

[Figure]

*Figure 2. Measured daily GPP (μmol m⁻² day⁻¹) as function of the stomatal O₃ flux (nmol m⁻²) accumulated over the five preceding days.*

**The referee's comment: 1.2** Furthermore, your comment "The increasing GPP involves forest recovery from acidification (Neirynck et al., 2008). The negative relation between GPP and $O_3$ dose for this period might wrongly be interpreted as an $O_3$ effect." is worrying, given the scope of your paper. You basically say that there is another parameter that has influenced the GPP development over time and has hence interacted with the ozone effect. Can you please describe how your methodological approach is able to disentangle the effect of a) recovery from acidification and b) ozone on GPP?

**The authors' response:**

Our methodological approach is able to disentangle the two effects because it involves the modelling of GPP from a set of input variables that excludes ozone but includes the main drivers of GPP - also accounting for the GPP increase as a recovery response from acidification. These drivers are, amongst others, the environmental drivers and LAI. Those together explain the bulk of daily and yearly variability in GPP.

**The referee's comment: 1.3** Also, line 457 of the paper "Overall, no significant $O_3$ effects on daily and growing season GPP accumulated over the growing season were found" does seem to counteract Figure 1 (as given in your recent response to reviewers), and this figure includes even measured as opposed to modelled data. You might want to briefly comment on this.

**The authors' response:**

We agree with the referee that this statement seems to contradict the information contained in Figure 1 (= Figure 3 A-C below). To be more correct, we will rephrase the line to "Overall, no significant $O_3$ effects on daily and growing season GPP accumulated over the growing season were found with our modelling approach." (Line 451).

We nevertheless strongly believe that Figure 1 with the plots of measured growing season GPP versus $F_{st}$, AOT40, and $POD_1$ is misleading in the sense that it suggests a strong $O_3$ effect on GPP, whereas based on our expert judgement we think the plots reflect more the fact that the period of increasing GPP by forest recovery from acidification happens to coincide with a period of decreasing $O_3$ concentrations. It is actually only with a modelling approach like the one we applied here in this study that these "confounding" effects can be disentangled.

We furthermore believe that the negative correlation at low stomatal $O_3$ fluxes is not causal, because the trends shown in Figure 1 (= Figure 3 A-C below) are not entirely what one would expect. For example, if we look at the plot of measured GPP versus $F_{st}$ (Figure 3 A), we see GPP decreasing with increasing $F_{st}$ in the low $F_{st}$ range (50 – 70 mmol $O_3$ m$^{-2}$) but we don't see a further GPP decline in the middle and high $F_{st}$ range (70-150 mmol $O_3$ m$^{-2}$). If $O_3$ would negatively affect GPP, we would a priori expect a decrease in GPP at high $F_{st}$ and much less at low $F_{st}$.

We added the figure below to the supplement of the manuscript (Figure S4) as suggested by the referee in comment 1.1 and added information about this figure at the appropriate text locations in the Results in paragraph 3.3 Ozone effects on GPP and in Discussion in paragraph 4.3 Ozone effects on GPP.

[Figure]

*Figure 3. (A, B, C) The measured growing season GPP as function of total stomatal O3 flux ($F_{st}$), AOT40, and $POD_1$. PLA = projected leaf area. (D, E, F, G) Time series of measured growing season GPP, $F_{st}$, AOT40, and $POD_1$.*

**The referee's comment:** 2. Former comment #6: I am fine with the response and don't have to see the results of the 1-day and 6-day delay period.

General comment: Line 794 and 819: Change "in function of" to "as a function of"

**The authors' response:**

We thank the referee for this comment and corrected this in the manuscript (Line 806 and 831).

We also thank the referee for his multitude of critical and constructive comments that helped us to improve our manuscript.

[revised manuscript text omitted]